# Dynamic modulation of activity in cerebellar nuclei neurons during pavlovian eyeblink conditioning in mice

Michiel M ten Brinke[1†], Shane A Heiney[2†], Xiaolu Wang[1†], Martina Proietti-Onori[1], Henk-Jan Boele[1], Jacob Bakermans[1], Javier F Medina[2*], Zhenyu Gao[1*], Chris I De Zeeuw[1,3]

[1]Department of Neuroscience, Erasmus Medical Center, Rotterdam, Netherlands; [2]Department of Neuroscience, Baylor College of Medicine, Houston, United States; [3]Netherlands Institute for Neuroscience, Royal Academy of Arts and Sciences (KNAW), Amsterdam, Netherlands

**Abstract** While research on the cerebellar cortex is crystallizing our understanding of its function in learning behavior, many questions surrounding its downstream targets remain. Here, we evaluate the dynamics of cerebellar interpositus nucleus (IpN) neurons over the course of Pavlovian eyeblink conditioning. A diverse range of learning-induced neuronal responses was observed, including increases and decreases in activity during the generation of conditioned blinks. Trial-by-trial correlational analysis and optogenetic manipulation demonstrate that facilitation in the IpN drives the eyelid movements. Adaptive facilitatory responses are often preceded by acquired transient inhibition of IpN activity that, based on latency and effect, appear to be driven by complex spikes in cerebellar cortical Purkinje cells. Likewise, during reflexive blinks to periocular stimulation, IpN cells show excitation-suppression patterns that suggest a contribution of climbing fibers and their collaterals. These findings highlight the integrative properties of subcortical neurons at the cerebellar output stage mediating conditioned behavior.
DOI: https://doi.org/10.7554/eLife.28132.001

*For correspondence:
jfmedina@bcm.edu (JFM);
z.gao@erasmusmc.nl (ZG)

†These authors contributed equally to this work

Competing interests: The authors declare that no competing interests exist.

## Introduction

The cerebellar cortex, like the neocortex, is well suited for establishing new associations required during memory formation. It is becoming increasingly clear that, like cortical and subcortical structures (e.g. *Constantinople and Bruno, 2013*; *Koralek et al., 2012*; *Igarashi et al., 2014*; *Douglas et al., 1995*; *Sherman and Guillery, 2004*), the cerebellar cortex and nuclei exhibit complex interplay, for instance through reciprocal nucleo-cortical projections (*Gao et al., 2016*). Classical Pavlovian eyeblink conditioning has proven an ideal model for studying the neural mechanisms underlying associative learning, and offers a suitable paradigm to address the question of how the cerebellar nuclei integrate their cerebellar cortical and extra-cerebellar input.

As a simple and quintessential behavioral manifestation of learning and memory, eyeblink conditioning depends on the cerebellar cortex and nuclei (*McCormick et al., 1982*; *McCormick and Thompson, 1984*; *Yeo et al., 1985a1985*, *1985b*), which are known to facilitate processes that require precise timing (*Ivry and Keele, 1989*; *Breska and Ivry, 2016*). In short, animals learn to respond to a neutral conditional stimulus (CS), such as a light, with a well-timed conditioned blink response (CR), when the CS is consistently paired at a fixed temporal interval with an unconditional blink-eliciting stimulus (US), such as a corneal air puff. Lobule HVI of the cerebellar cortex and its downstream target, the interposed nucleus (IpN), are essential for the manifestation of this conditioned eyelid behavior (*McCormick et al., 1982*; *McCormick and Thompson, 1984*; *Yeo et al.,*

*1985a1985*, *1985b*; *Clark et al., 1992*; *Krupa and Thompson, 1997*; *Ohyama et al., 2006*; *Mostofi et al., 2010*; *Heiney et al., 2014b*). Genetic and pharmacological manipulations of cerebellar cortical Purkinje cells indicate that the expression of CRs may require various cell physiological processes, including plasticity at the excitatory parallel fiber to Purkinje cell synapse (*Ito and Kano, 1982*; *Koekkoek et al., 2003*; *Schonewille et al., 2010*), inhibition at the molecular layer interneuron to Purkinje cell synapse (*ten Brinke et al., 2015*), as well as intrinsic mGluR7-mediated processes in Purkinje cells (*Johansson et al., 2015*). IpN neurons that can drive eyeblink behavior through the red nucleus and facial nucleus provide a feedback to the cerebellar cortex that amplifies the CR (*Gao et al., 2016*; *Giovannucci et al., 2017*). Moreover, IpN neurons receive excitatory inputs from mossy fiber and climbing fiber collaterals, which need to be integrated with the inhibitory inputs from the Purkinje cells and local interneurons and possibly even from recurrent collaterals of the nucleo-olivary neurons (*Chan-Palay, 1973*, *Chan-Palay, 1977*; *de Zeeuw et al., 1988*, *1997*; *Van der Want et al., 1989*; *Uusisaari et al., 2007*; *Uusisaari and Knöpfel, 2008*, *2011*, *2012*; *Bagnall et al., 2009*; *Kodama et al., 2012*; *Witter et al., 2013*; *Boele et al., 2013*; *Najac and Raman, 2015*; *Canto et al., 2016*).

IpN neurons and their inputs are endowed with ample forms of plasticity, all of which may in principle be implicated in eyeblink conditioning (*Ohyama et al., 2006*; *Foscarin et al., 2011*; *Zheng and Raman, 2010*). These include for example short-term and long-term potentiation (LTP) at the mossy fiber to IpN neuron synapse (*Person and Raman, 2010*), both of which may facilitate the induction of intrinsic plasticity (*Aizenman and Linden, 1999* and *Aizenman and Linden, 2000*; *Zheng and Raman, 2010*), and learning-dependent structural outgrowth of mossy fiber collaterals in the cerebellar nuclei, which may be directly correlated to the rate and amplitude of CRs (*Boele et al., 2013*). So far, extracellular recordings of neurons in the cerebellar nuclei have revealed CR-related increases in activity that lead the eyeblink movement (anterior IpN; *McCormick and Thompson, 1984*; *Berthier and Moore, 1990*; *Gould and Steinmetz, 1996*; *Choi and Moore, 2003*; *Halverson et al., 2010*; *Heiney et al., 2014b*), and CR-related increases and decreases in activity that lag the eyeblink movement (posterior IpN; *Gruart et al., 2000*; *Delgado-García and Gruart, 2005*; *Sánchez-Campusano et al., 2007*). However, comprehensive trial-by-trial analysis of different components of IpN modulation in relation to conditioned behavior, as well as evaluation against conditioning-related modulation of activity in the cerebellar cortex and other non-cortical inputs, is lacking.

Here, we recorded conditioning-related activity in an identified blink area of the anterior interpositus nucleus of awake behaving mice, and used in-depth trial-by-trial correlational analysis, optogenetic manipulation and computationally modeled IpN output based on previously reported eyelid-related Purkinje cell modulation (*ten Brinke et al., 2015*) to detail and cement the causal role of IpN activity in conditioned behavior. Our results reveal an intricate dynamic modulation of cerebellar nuclei activity during the generation of conditioned movements. We discuss how these findings shed light on a number of electrophysiological properties of the olivocerebellar network as integrated at the IpN in the normal functional context of eyeblink conditioning in awake, behaving mice. Together, our results highlight how neurons in the IpN integrate cortical and non-cortical input to establish a circuitry optimally designed to generate well-timed conditioned motor responses.

## Results

### Characterization of cerebellar IpN neurons

To explore the characteristics of IpN spike modulation and its relation to conditioned eyelid behavior, we made extracellular recordings across 19 mice that were trained or in training. During recordings, the mice were head-fixed on top of a cylindrical treadmill, fully awake and able to show normal eyelid behavior in response to the experimental stimuli, which included a 260 ms green LED light (CS) that co-terminated with a 10 ms corneal air puff (US), yielding a 250 ms CS-US interval (*Figure 1A*).

Initially, we collected 270 recordings of IpN cells that were located below Purkinje cell layers at depths between 1500–3000 μm (*Figure 1B*), that were recorded during at least 10 valid CS-US trials, ánd that showed modulation in their firing rate in response to the CS and/or US. Firing frequency was similar along the depth of the IpN (r = 0.085, p=0.16, n = 270, Pearson), averaging 67 ± 27 Hz.

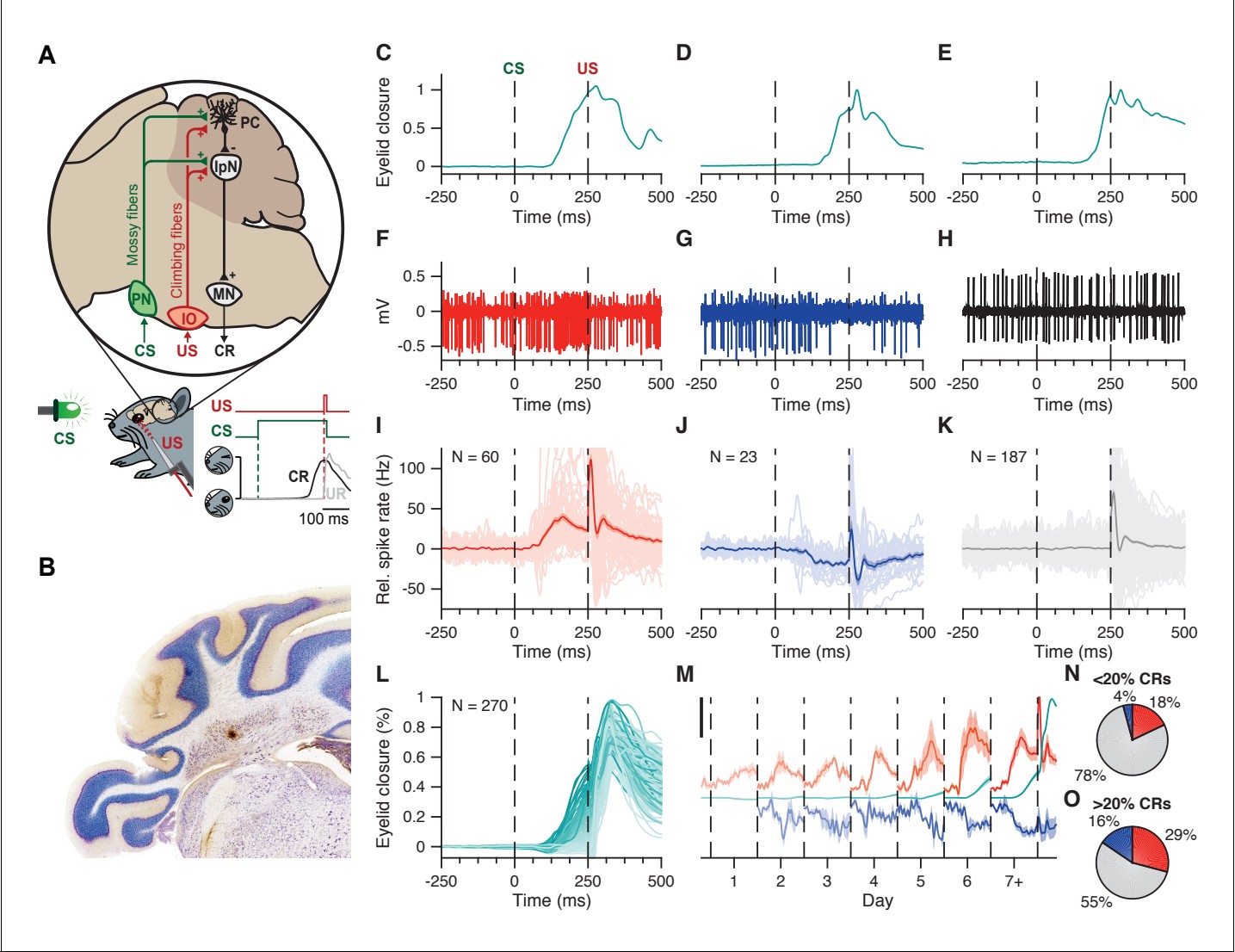

**Figure 1.** Interpositus nucleus electrophysiology overview. (A) CS and US signals are transmitted through mossy fibers and climbing fibers, respectively, to Purkinje cells in the simplex lobule (HVI) of the cerebellar cortex, and to the IpN, to which cortical Purkinje cells send their inhibitory projection. Combined disinhibition and excitation then lead the IpN to drive CRs. Paired trials consisted of a 260 ms LED light CS, co-terminating with a 10 ms corneal air puff. (B) Coronal cerebellar section showing a typical recording site in the IpN. C-E Example eyelid traces for trials where the recorded cell showed spike facilitation (C), suppression (D), or no modulation (E) in the CS-US interval. (F-H) Electrophysiological trace showing IpN activity corresponding to the example eyelid traces in (C-E). (I) Spike trace averages for 60 cells from the first dataset showing at least 5 Hz facilitation in the CS-US interval; grand average is shown with SEM. (J) Same as in I, but for 23 suppressive cells. (K) Same as in I, but for 187 non-modulating cells. (L) Average traces of eyelid behavior for all 270 cells in the first dataset, color coded for average CR amplitude at US onset. (M) Daily CS-US interval averages of facilitatory cells (top traces, orange), suppressive cells (bottom traces, blue), and eyelid behavior for all cells showing modulation (middle traces, teal). Black scale bar denotes 25 Hz/% eyelid closure. (N, O) Proportion of cells showing facilitation, suppression, or no modulation, across cells with <20% CRs (n = 167), and cells with >20% CRs (n = 103). IpN: interpositus nucleus; IO: inferior olive; MN: motor nucleus; PC: Purkinje cell; PN: pontine nuclei.

DOI: https://doi.org/10.7554/eLife.28132.002

The following figure supplement is available for figure 1:

**Figure supplement 1.** Overview of IpN recording datasets.

DOI: https://doi.org/10.7554/eLife.28132.003

Within the entire dataset, 60 cells showed at least 5 Hz facilitation in the CS-US interval (22.2%, *Figure 1C,F,I*), and 23 cells showed at least 5 Hz suppression (8.5%, *Figure 1D,G,J*), leaving 187 cells that did not show any clear spike rate modulation within the CS-US interval (69.3%, *Figure 1E, H,K*; see Materials and methods).

In terms of eyelid behavior across the dataset (*Figure 1L*), CRs were present in at least 20% of trials in 103 cells (38%); this is the criterion used throughout the paper when referring to recordings with behavior. The magnitude of spike facilitation showed a steady significant increase over the course of conditioning (r = 0.413, p=0.001, n = 60; Spearman); spike suppression was similarly inclined (r = 0.354, p=0.0972, n = 23; Spearman; *Figure 1M*). Although both types of spike rate modulation manifested before conditioned behavior did (*Figure 1M*), they were more prevalent in recordings with conditioned behavior (*Figure 1O*) than in those without (*Figure 1N*; facilitation: 29.1% (30/103) vs. 18% (30/167), p=0.0463; suppression: 15.5% (16/103) vs. 4.2% (7/167), p=0.0025, Chi-square test).

To corroborate the main findings of the first dataset, an independently obtained, second dataset of 102 IpN recordings was analyzed in the same way. The collection of this dataset was optimized in that: (i) rather than over the course of conditioning, these recordings were obtained after thorough training, and only from mice that showed at least 80% CRs; (ii) the recordings were all located no more than ~300 μm away from IpN sites confirmed to be eyelid-related through micro-stimulation (see Materials and methods, and *Heiney et al., 2014b*); and (iii) the CS was 220 ms, the US was 20 ms, resulting in a CS-US interval of 200 ms. Thus, rather than focusing on the course of training and across-trial variability, the recordings of the second dataset served to capture cerebellar nuclear dynamics in an optimally and fully conditioned system (for an overview of the two different datasets, see *Figure 1—figure supplement 1*).

In subsequent analyses, we focused on three relevant time intervals during stimulus presentation: the last 200 ms of the CS-US interval, during which broad Purkinje cell simple spike modulation takes place; 50–130 ms after the CS, during which CS-related climbing fiber activity emerges (*Ohmae and Medina, 2015*; *ten Brinke et al., 2015*); and the first 60 ms after the US, during which robust climbing fiber signals take place (the data structure in *Source data 1* contains these variables for both datasets).

## IpN facilitation but not suppression can drive conditioned eyelid behavior

We first sought to characterize whether facilitation and suppression observed in IpN neurons during the last 200 ms of the CS-US interval correlates with the amplitude of eyelid closure (*Figure 2*). Across the first dataset, there were 60 cells showing CS-US facilitation, with durations of 104 ± 41 ms, and magnitudes averaging 22.5 ± 20.2 Hz and correlating to both percentage CRs (r = 0.534, p<0.0001, n = 60, Spearman) and CR amplitude (r = 0.52, p<0.0001, Spearman). Out of 30 facilitation cells recorded in the presence of CRs (*Figure 2A*), 17 showed significant positive trial-by-trial correlations between firing rate and CR amplitude (57%; p=0.0001, bootstrap with 500 repetitions; *Figure 2B*), whereas none showed negative correlations. A linear mixed model with random intercepts and slopes for these significant cells predicts 0.49% more eyelid closure at US onset per unit increase of spike facilitation in Hz (*Figure 2B*, black dotted line; p<0.0001, *Supplementary file 1*). Exploration of trial-by-trial spike-eyelid correlations across a correlation matrix reveals their temporal distribution, as explained previously (*ten Brinke et al., 2015*). Here, correlations within 20 ms windows, taken at 10 ms steps across the trial timespan, show at which spike times and which eyelid times the strongest spike-eyelid correlations exist, both when taking eyelid position (*Figure 2C*) and eyelid velocity (*Figure 2—figure supplement 1A*) as the lead parameter. A cluster of markedly positive correlations can be seen in the upper triangle of the matrix in the second half of the CS-US interval, running parallel to the diagonal. This means the strongest across-trial correlations occurred between spiking and subsequent behavior. Indeed, temporal cross-correlations between average spike and eyelid traces confirmed that spike facilitation fits best with subsequent eyelid behavior, averaging 25 ± 25.6 ms (p=0.0001, Wilcoxon signed rank test; *Figure 2D*).

The character and predictive power of CS-US facilitation in the IpN was corroborated by the second dataset (*Figure 2E–H*), with 70 cells showing facilitation (59.2 ± 38 Hz) in the presence of conditioned behavior (*Figure 2E*), and 49 of these cells showing significant positive correlations (70%). A linear mixed model with random intercepts and slopes predicted 0.64% more eyelid closure at US

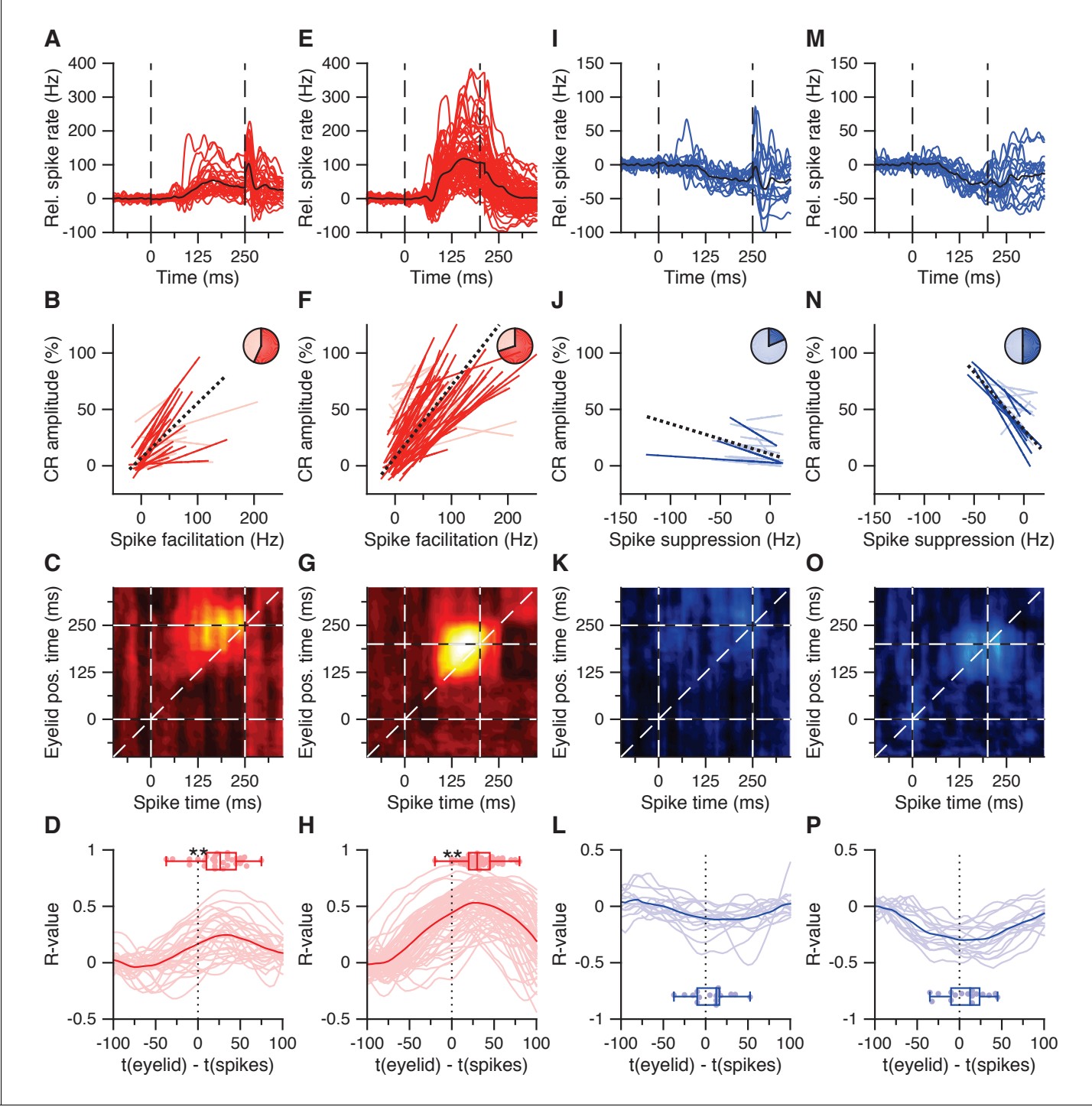

**Figure 2.** CS-US interval IpN modulation relates to eyelid behavior. (**A**) Average spike traces of 30 facilitatory neurons in the first dataset recorded with conditioned behavior. Black trace denotes average. Time is shown relative to CS onset. (**B**) Trial-by-trial spike-eyelid correlation lines for the cells in A, with plain red lines showing significant correlations, light red lines showing the non-significant ones, and the pie chart showing their proportionality. (**C**) Average correlation matrix showing the average temporal distribution of the spike-eyelid position correlations of the cells in A. R-values counter to the correlational direction of interest were nullified before averaging. (**D**) Temporal cross-correlations for the cells in A (light red) and their mean (plain red). Box plot shows the time of maximum correlation for all cells. (**E-H**) Same as in (**A-D**), here for 70 facilitation cells with conditioned behavior from the independent second dataset. (**I-L**) Same as in (**A-D**), here for 16 suppression cells recorded with conditioned behavior from the first dataset. (**M-P**) Same as in (**I-L**), here for 16 suppression cells recorded with conditioned behavior from the second dataset.

DOI: https://doi.org/10.7554/eLife.28132.004

*Figure 2 continued on next page*

*Figure 2 continued*

The following figure supplement is available for figure 2:

**Figure supplement 1.** Spike-eyelid velocity correlation matrices.

DOI: https://doi.org/10.7554/eLife.28132.005

onset per 1 Hz increase in facilitation (p<0.0001, *Supplementary file 1*). The correlation matrices relating spike activity to eyelid position (*Figure 2G*) and velocity (*Figure 2—figure supplement 1B*) showed hotspots of strong correlation hovering above the diagonal in the CS-US interval, with temporal cross-correlations again confirming the best fit between spike facilitation and eyelid behavior to be at significantly positive offsets (33.4 ± 19.3 ms, p<0.0001, Wilcoxon signed rank test; *Figure 2H*).

In the first dataset, CS-US suppression was less prevalent than facilitation. Within the 23 cells (8.5%) showing an average drop of at least 5 Hz in the CS-US interval (11.2 ± 6 Hz, duration: 70 ± 44 ms), the magnitude of suppression correlated to both CR percentage (r = 0.526, p=0.01, n = 23, Spearman) and CR amplitude at US onset (r = 0.443, p=0.0356, n = 23, Spearman). Additionally, cells with stronger suppression showed higher spike irregularity (i.e., higher CVs: r = 0.587, p=0.0038, Spearman). Compared to facilitation, suppression had lower amplitudes (baseline z-score: 5.3 ± 1.5 vs 7.6 ± 4.7, p=0.0007, Mann-Whitney U test, MWU) and lower durations (53 ± 38.4 vs 95.5 ± 36.2 ms, p=0.0012, MWU), but similar onset (132 ± 21.2 vs 136.5 ± 19.9 ms, p=0.6, MWU) and modulation peak times (176 ± 37.2 vs 166 ± 34.2 ms, p=0.6, MWU). Relative spike suppression was further correlated to eyelid closure at US-onset relative to baseline for all recordings of suppressive cells with conditioned behavior (n = 16, *Figure 2I*). Of these cells, three showed significant negative trial-by-trial correlations between firing rate and CR amplitude (19%, p<0.0001, bootstrap with 500 repetitions; *Figure 2J*), whereas none showed positive correlations. A linear mixed model with random intercepts and slopes for the three significant cells estimated 0.27% more eyelid closure at US onset per unitary increase of suppression in Hz (p=0.0272, *Supplementary file 1*). The correlation matrix showed no particular concentration of the spike-eyelid correlations of suppressive cells with conditioned behavior within the CS-US interval (position: *Figure 2K*; velocity: *Figure 2—figure supplement 1C*), and temporal cross-correlation showed a best fit between average spike and eyelid traces at very small temporal offsets that were on average close to zero (6.4 ± 23 ms, p=0.2336, Wilcoxon signed rank test; *Figure 2L*).

In the second dataset, 16 cells showed CS-US spike suppression (12.1 ± 5 Hz) in the presence of conditioned behavior (*Figure 2M*), with 8 cells showing a significant negative relationship between relative firing rate and CR amplitude at US onset (p<0.0001, bootstrap with 500 repetitions; *Figure 2N*) and a linear mixed model estimating 1.01% more eyelid closure per 1 Hz deeper suppression (p<0.0001, *Supplementary file 1*). Although the second dataset showed stronger facilitation than the first dataset (59.2 ± 38.1 vs 30.1 ± 25.3 Hz, p=0.0009, MWU), arguably due to superior CR performance (mean percentage CRs: 86.2 ± 17.3%; mean CR amplitude: 43.5 ± 17% eyelid closure), the level of suppression was not different between the two datasets (11.9 ± 5 Hz vs 12.9 ± 6.4 Hz, p=0.864, MWU). Moreover, the correlation matrices for eyelid position (*Figure 2O*) and velocity (*Figure 2—figure supplement 1D*) echo-ed the finding that the time difference between suppressive modulation and eyelid movement was not significantly different from zero on average (6.9 ± 25 ms, p=0.2439, Wilcoxon signed rank test; *Figure 2P*).

Together, these results, found across two independent datasets, reveal a large portion of IpN cells showing spike facilitation during the CS-US interval that predicts subsequent conditioned eyelid behavior. In contrast, there is a smaller number of suppressive neurons that are unlikely to drive eyelid movements, because their activity is less correlated with CR amplitude and often lags the eyelid movement (or leads it by an insufficient amount).

## Optogenetic inhibition of IpN eliminates CRs

If CS-US facilitation in the IpN indeed controls eyelid closure, one would assume that inhibiting IpN activity during the CS-US interval is sufficient to abolish CRs. Pharmacological interventions suggest this is indeed so, with muscimol and/or lidocaine injections in the anterior IpN eliminating CRs completely (rabbit: *Bracha et al., 1994*; *Ohyama et al., 2006*; rat: *Freeman et al., 2005*; mouse:

*Heiney et al., 2014b*). To further establish the causal role of IpN facilitation in the expression of specifically conditioned eyelid behavior, we employed optogenetic inhibition of the IpN in a set of three fully trained mice. Optic fibers were implanted near the IpN of L7cre-Ai27 mice, in which ChR2 was expressed in all Purkinje cells (see Materials and methods). During optogenetic excitation of Purkinje cell axon terminals, IpN neurons suppress their firing activity (*Witter et al., 2013*; *Canto et al., 2016*; *Figure 3A*). If excitatory rather than suppressive modulation in the IpN plays a causal role in driving the CRs, CRs should be selectively eliminated by optogenetic stimulation. Indeed, the conditioned behavior acquired by all three mice was readily disabled in virtually all trials randomly chosen to include optogenetic stimulation (all p<0.0001, MWU; *Figure 3B,C*). This result expands on the observation that IpN facilitation can elicit eyeblinks (*Hesslow, 1994b*; *Heiney et al., 2014b*), by showing that it is decidedly necessary for the expression of conditioned eyelid responses.

## CS-related transient spike pauses and subsequent rapid excitation in IpN neurons

Next, we examined whether and how previously reported CS-related complex spikes driven by climbing fibers in Purkinje cells (*Ohmae and Medina, 2015*; *ten Brinke et al., 2015*) may be reflected in IpN activity. Although IpN cells do not show an identifiably distinct spike type upon climbing fiber activation, previous work characterized a cerebellar nuclear equivalent to synchronous complex spike activity as a transient suppression in spike rate (e.g. *Hoebeek et al., 2010*; *Bengtsson et al., 2011*; *Witter et al., 2013*; *Bengtsson and Jörntell, 2014*; *Tang et al., 2016*). Based on the latency of CS-related complex spikes in eyelid-related Purkinje cells (*Figure 4A,D*), we looked at rapid spike deviations exceeding 4.5 baseline SDs between 50–125 ms post-CS in IpN neurons. Seven cells showed a striking transient suppression at 88 ± 12 ms (*Figure 4B,E*), which we refer to as a CS pause, with five of them showing a subsequent facilitation in the CS-US interval (71.4%, vs 55/263 (21%), p=0.0006, Fisher's test).

The presence and character of CS pauses as observed in the first dataset were strongly reflected in the second dataset, which was obtained from fully trained, high-performing mice. Out of 102 cells,

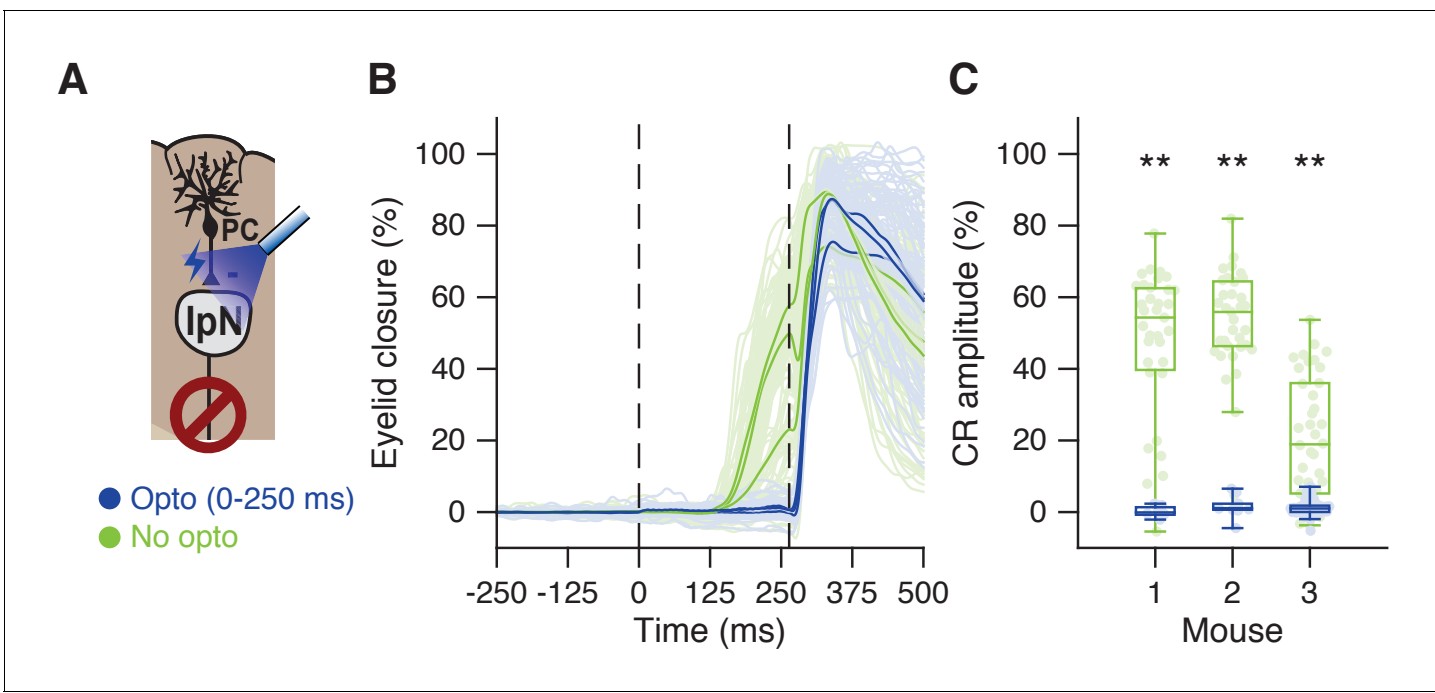

**Figure 3.** Optogenetic prevention of IpN facilitation eliminates CRs. (**A**) In L7cre-Ai27 mice, Purkinje cells in the simplex lobule that express ChR2 exert a powerful inhibitory influence on the IpN upon optogenetic stimulation (see *Witter et al., 2013*; *Canto et al., 2016*). (**B**) Average eyelid traces for trials with (dark blue) or without (green) optogenetic stimulation throughout the CS-US interval, for three mice. C Eyelid closure at US onset for the data shown in B, separated by mouse.

DOI: https://doi.org/10.7554/eLife.28132.006

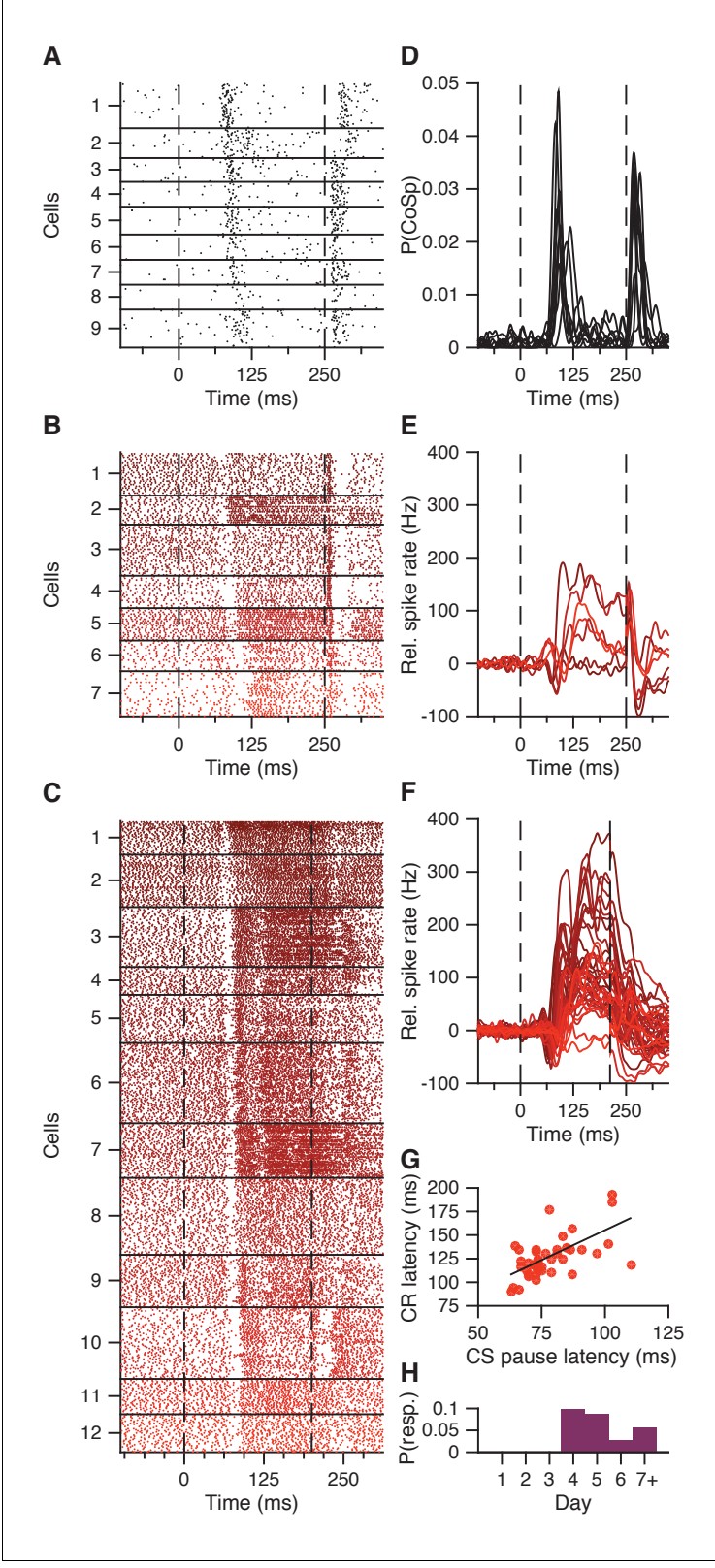

**Figure 4.** CS pause response reflecting CS-related Purkinje cell complex spike. (A) Combined complex spike raster plot for 9 Purkinje cells ordered by the latency of their clear CS-related complex spike response in addition to the US-related complex spike (data from *ten Brinke et al., 2015*). (B) Combined raster plot for 7 IpN neurons, ordered by the latency of their CS pause in spike activity. (C) Same as in B, but for 12 of the 41 IpN neurons in the
*Figure 4 continued on next page*

*Figure 4 continued*

second dataset that showed a CS pause. (D-F) Average spike traces corresponding to the cells shown in (A-C), with (D) showing the probability of a complex spike instead of relative spike rate. (G) CS pause latency plotted against the latency at which the CR passes 5% eyelid closure, for all 45 cells showing both properties across the original and the second dataset. (H) Probability across days of finding a transient spike response at CS-complex spike latency, inhibitory (this figure) or excitatory (*Figure 4—figure supplement 1*), in the first dataset.

DOI: https://doi.org/10.7554/eLife.28132.007

The following figure supplement is available for figure 4:

**Figure supplement 1.** Transient spike increase at CS-complex spike latency.

DOI: https://doi.org/10.7554/eLife.28132.008

41 showed CS pauses (*Figure 4C,F*), and here too cells with a CS pause were more likely to show CS-US facilitation than those without (39/41 (95%) vs 31/61 (51%) cells, p<0.0001, Fisher's test). Importantly, there was a clear across-cell correlation between average CS pause latency and CR latency, whether looking at both datasets combined (n = 45, r = 0.582, p<0.0001, Pearson; *Figure 4G*), or just the second dataset in isolation (n = 41, r = 0.403, p=0.0089, Pearson).

Interestingly, there were four cells in the first dataset that showed a transient increase, rather than a decrease, in firing at the same latency as the CS pause (*Figure 4—figure supplement 1A,B*). Rather than showing subsequent facilitation, two of these cells showed CS-US suppression (50%, vs 21/266 (7.9%) in the other cells, p=0.0021, Fisher's test). The suppression did not seem more enhanced in these cells compared to other suppression cells (all p>0.25; *Figure 4—figure supplement 1C,D*). Moreover, this transient increase in firing was not observed in any of the cells in the second dataset, which was specifically focused on eyelid-related IpN areas as identified by microstimulation-driven blinks (see Materials and methods).

Across the first dataset, which was obtained over the course of conditioning, none of the 11 cells showing a transient spike response (CS pause or a transient increase in spikes) at the CS-related complex spike latency were recorded before day 4 of training (day 1–3, 0/99 cells; day 4+, 11/171, p=0.0058, Fisher's test; *Figure 4H*), which is consistent with the notion that CS-related climbing fiber response were acquired during learning (*Ohmae and Medina, 2015*; *ten Brinke et al., 2015*).

To further study the potential impacts of the climbing fiber inputs on CN modulation during CS and US, we compared the activity patterns of neurons with CS pauses to those without (*Figure 5A, B,F,G*). We found that cells with CS pauses not only have larger facilitation amplitudes compared to those without, but also showed a markedly more rapid excitatory spike profile right after the CS pause latency. In the first dataset, they showed significantly higher CR-related increases in firing rate during the CS-US interval (41.5 ± 33.1 vs 15.2 ± 8.9 Hz, p=0.0013, MWU; *Figure 5C*), and they also showed distinctly higher maximum spike rate velocities (4.9 ± 1.6 vs 1.4 ± 0.5 Hz/ms, p=0.0002, MWU; *Figure 5D*). While in the second dataset maximum facilitation was not significantly higher in facilitation cells with a CS pause (65.1 ± 39.5 Hz) compared to those without (51.7 ± 35.4 Hz, p=0.1594, MWU; *Figure 5H*), the former group did show significantly higher maximum spike rate velocities (5.6 ± 2.5 vs 3.6 ± 1.5 Hz/ms, p=0.0004, MWU; *Figure 5I*). Note that CS pause cells tended to be recorded less deeply in the nuclei than non-CS pause cells, in both the first (1840 ± 320 vs 2130 ± 350 µm, p=0.0298, MWU; *Figure 5E*) and second dataset (2350 ± 140 vs 2450 ± 211 µm, p=0.0372, MWU; *Figure 5J*).

To find out to what extent the CS-US facilitation in the IpN could result from merely Purkinje cell simple spike suppression (without taking the complex spike responses or collateral input to IpN neurons into account), we used simple spike profiles of 26 Purkinje cells reported previously (*ten Brinke et al., 2015*), to model spike modulation for as many IpN neurons (*Figure 5—figure supplement 1A,B*), using model parameters based on *Yamazaki and Tanaka, 2007* and *Person and Raman (2011)*; *Figure 5—source data 1*). The modeled CS-US facilitation resulting from averaging the simple spike activity of all Purkinje cells together aligned better with the recorded CS-US facilitation cells without a CS pause than those with CS pause, in the first dataset (*Figure 5—figure supplement 1C–H*). The model could also reproduce the much stronger and rapid excitatory response in the IpN neurons with a CS pause, but only if these IpN neurons were assumed to receive input from the Purkinje cells whose activity was suppressed the most (*Figure 5—figure supplement 1A,B,D*, blue traces)

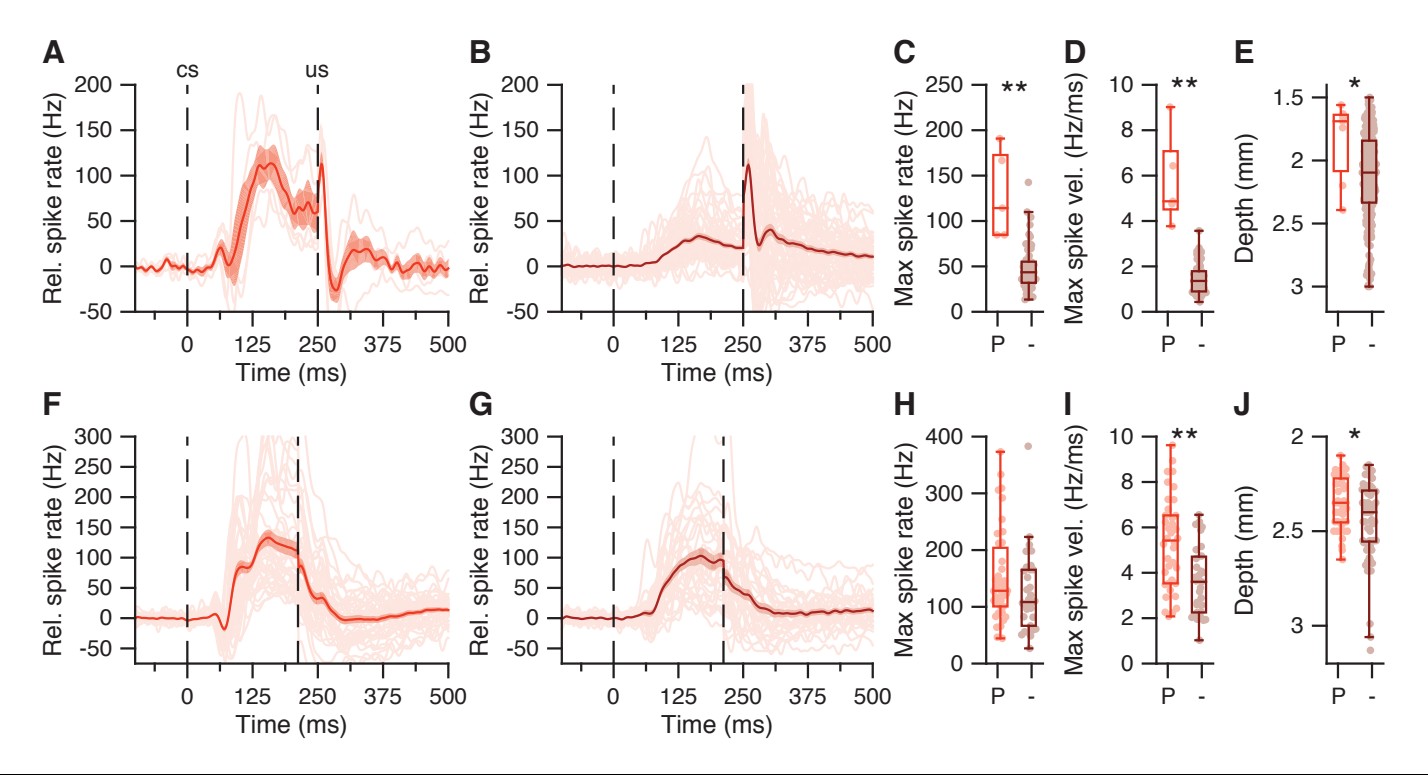

**Figure 5.** CS-US facilitation in IpN cells with and without CS pause. (**A**) Average spike traces for 5 IpN cells showing CS-US facilitation and a CS pause. Grand average shown in red, with SEM. (**B**) Same as in A, here showing 55 IpN cells with CS-US facilitation but without a CS pause. (**C**) Maximum spike rate in the CS-US interval for facilitation cells with CS pause (P) and without CS pause (-), for the data in A and B. (**D**) Same as C, here showing maximum spike rate velocity. (**E**) Same as in C, here showing recording depth. (**F-J**) Same as in (**A-E**), here showing data from the second dataset; 39 IpN cells showed CS-US facilitation and a CS pause, and 31 cells showed CS-US facilitation but not a CS pause.

DOI: https://doi.org/10.7554/eLife.28132.009

The following source data and figure supplement are available for figure 5:

**Source data 1.** DCN model properties.
DOI: https://doi.org/10.7554/eLife.28132.011
**Figure supplement 1.** Modeled IpN modulation.
DOI: https://doi.org/10.7554/eLife.28132.010

Together, these findings establish a clear reflection in the IpN of the CS-related climbing fiber response as a transient spike suppression and support its selective and acquired nature. Moreover, the link between CS pauses and the subsequent high and/or fast facilitation rates in IpN neurons across the datasets is suggestive of a potential functional role of CS-related climbing fiber activity in triggering strong excitation in the IpN, a possibility that is further explored below.

## IpN responses to the US

Central to the eyeblink conditioning paradigm is the notion that the US activates climbing fibers and evokes complex spikes in eyelid-related Purkinje cells, in particular if the mouse fails to make a conditioned response (*Ohmae and Medina, 2015*; *ten Brinke et al., 2015*). We therefore analyzed the spike modulation during the US epoch and searched for the modulation pattern indicative for an imprint of the US evoked complex spike activation. Interestingly, 10.6 ± 4.7 ms after US onset we found a marked increase in IpN spikes (99 ± 59 Hz), which exceeded five baseline SDs in 211 cells of the first dataset (78.2%, *Figure 6A,B*, see Materials and methods). This IpN response, here referred to as a US peak, had two important features: First, US peak amplitude, measured from baseline, diminished over the course of training, as apparent from a comparison of recordings made in the first four training days and those made subsequently (88 ± 48 vs. 29 ± 44 Hz, p<0.0001, MWU; *Figure 6C*). Second, US peak amplitude correlated negatively with conditioned behavior, in terms of

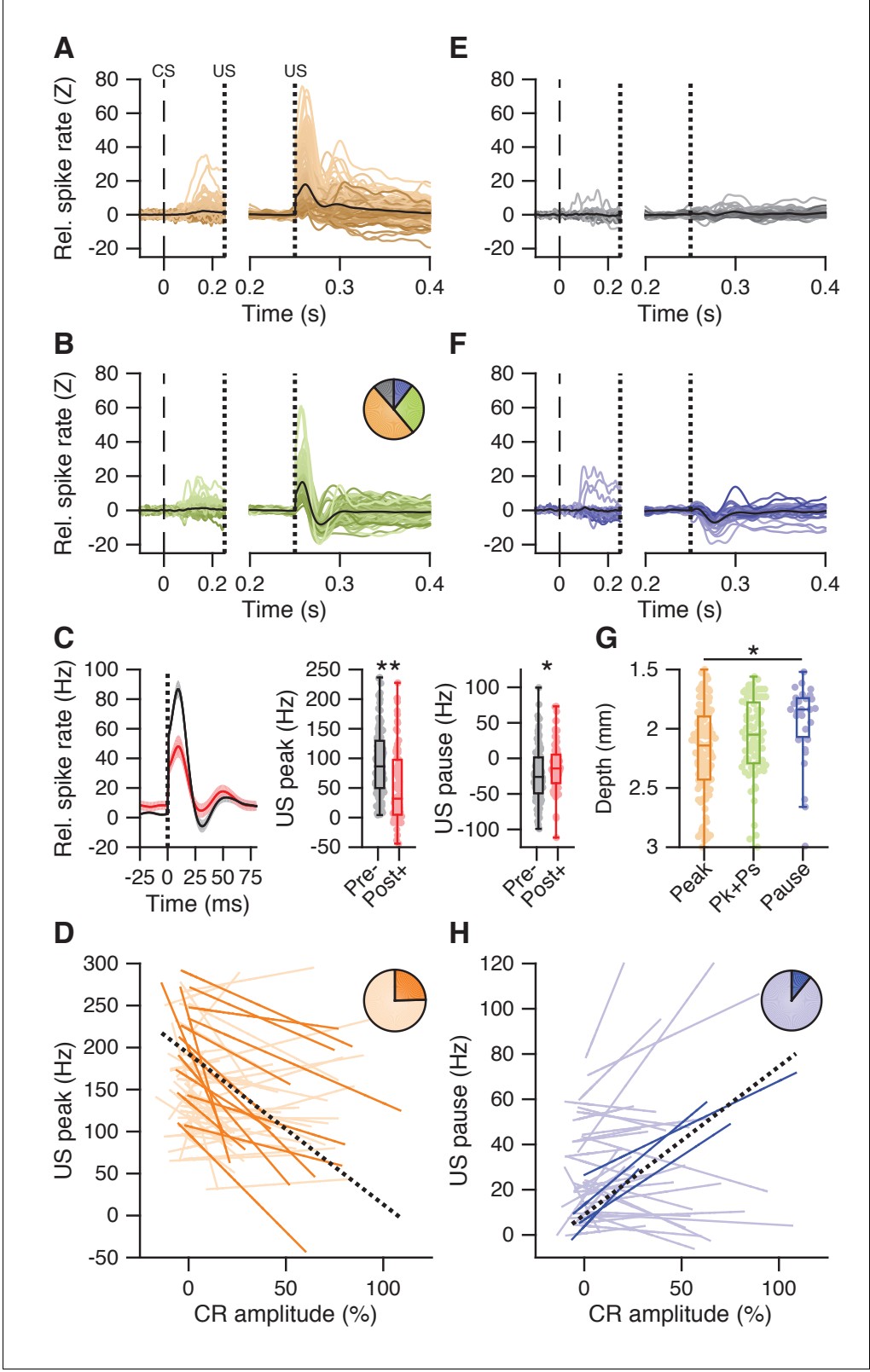

**Figure 6.** US peak and pause responses. (**A**) Average spike traces for cells showing a US peak but not a US pause response (n = 134); left panel shows activity in the CS-US interval aligned to baseline, right panel shows post-US activity aligned to the last 50 ms of the CS-US interval. (**B**) Same as in A, but for cells showing both a US peak and a US pause (n = 77). The pie chart shows the proportion of cells with a US peak (yellow), a US pause (blue), both

*Figure 6 continued on next page*

*Figure 6 continued*

(green), or neither (gray). (**C**) Left panel: average traces with SEM of spike activity relative to baseline for all cells recorded without conditioned behavior on days 1–4 (black), and all cells recorded after day four and with conditioned behavior (red). Middle panel: US peak amplitude relative to baseline was significantly higher in early recordings without conditioned behavior (pre-) than in later recordings with behavior (post+; p<0.0001). Right panel: same as the middle panel, here showing minimum firing rate in the US pause window relative to the last 50 ms in the CS-US interval; early recordings without CR behavior show lower values than later recordings with behavior (p=0.0312). (**D**) Significant (plain orange) and non-significant (light orange) trial-by-trial correlation lines for all cells showing conditioned behavior and a US peak (n = 62). Black dotted line shows a fit from a linear mixed model incorporating the significant cells. The pie chart inset shows the proportion of cells that were significant. (**E**) Same as in A, but for cells without a clear post-US response (n = 31). (**F**) Same as in A, but for cells with only a US pause response (n = 28). (**G**) Recording depth was different between cells separated on US responses, getting progressively less deep from cells with only a US peak (yellow, 2182 ± 373 µm), through cells showing both US peak and pause (green, 2075 ± 322 µm), to cells showing only a US pause (blue, 1939 ± 299 µm; p=0.0025). (**H**) Similar to D, but for all cells showing conditioned behavior and a US pause (n = 37). Note that the y-axis shows the firing rate during the US pause period, with higher values implying less pronounced pauses.

DOI: https://doi.org/10.7554/eLife.28132.012

The following figure supplement is available for figure 6:

**Figure supplement 1.** Eyelid opening in the CS-US interval relates to reduced US peak and absence of US-complex spike.

DOI: https://doi.org/10.7554/eLife.28132.013

---

both CR percentage (r = −0.426, p<0.0001, n = 270, Spearman) and CR amplitude at US onset (r = −0.333, p<0.0001, n = 270, Spearman), confirming its reduction as conditioned behavior is acquired. Furthermore, among 62 recordings with a US peak and CR behavior, trial-by-trial correlations revealed 13 cells in which US peak amplitude correlated negatively to CR amplitude (p<0.0001, bootstrap with 500 repetitions; *Figure 6D*). These significant recordings were twice as likely as the other 49 cells to also show significant positive correlations between CR amplitude and CS-US facilitation (69.2% (9/13) versus 36.7% (18/49), p=0.0077, Fisher test), which itself was three times as likely to occur in cells with US peak compared to those without (26.1% (55/211) vs. 5/59, 8.5% (5/59), p=0.007, Fisher test). Together, these results establish a substantial prevalence of US peak responses in the IpN, demonstrate their co-occurrence with conditioning-related facilitation and spike-eyelid correlations in the CS-US interval, and show cell- and trial-wide conditioning-related dynamics that are similar to those of US-related climbing fiber responses.

Interestingly, another relationship between the US peak and US-related climbing fiber responses was observed in a subgroup of neurons. Occasionally, mice tended to open their eye relative to baseline in response to the CS. Among 149 of the recordings with a US peak but without CR behavior, 17 showed positive across-trial correlations between eyelid amplitude and the US peak response (11.4%, p<0.0001, bootstrap with 500 repetitions; *Figure 6—figure supplement 1A*), i.e. more eyelid opening related to weaker US peaks. We analyzed the Purkinje cell data from *ten Brinke et al. (2015)* to see if eyelid opening responses were related to US-complex spikes. Indeed, across trials where mice opened their eye further in the CS-US interval, those without a US-related complex spike showed slightly more eyelid opening (−2.65 ± 1.85%) than those with a US-complex spike (−0.84 ± 1.86%, p=0.0254, MWU; *Figure 6—figure supplement 1B,C*). Thus, this additional property seems to link the US peak to US-related climbing fiber activity.

In addition to US peak responses, transient suppressive responses were also observed at 28.2 ± 4.9 ms post-US in 105 IpN cells (39%), with mean magnitudes of −50 ± 24 Hz (*Figure 6F*). The latency of this response, referred to as the US pause, fits with the idea of climbing fiber input inhibiting the cerebellar nuclei indirectly via Purkinje cell complex spikes (e.g. *Hoebeek et al., 2010*; *Lu et al., 2016*). Consistent with this notion, the majority of cells that showed a CS pause also showed a US pause (86%, vs. 105/263 (40%), p=0.0012, Fisher's exact test). We found considerable overlap in the prevalence of US peaks and pauses: a third of the cells showing the one response also showed the other (77 out of 239, 32.2%; *Figure 6B*). Additionally, cells with only a US pause tended to be more dorsally located (1939 ± 299 µm) than cells with only a US peak (2182 ± 373 µm), and cells that showed both responses averaged an intermediate depth (2075 ± 322 µm; p=0.0025, Kruskal-Wallis; *Figure 6G*). In terms of conditioning-related dynamics, the US pause showed

characteristics that were similar to those of US peaks. Across the first four training days, 47.1% of cells (66/140) showed a US pause, compared to 30% (39/130) on subsequent training days in the first dataset (p=0.0057, Chi-square test). Moreover, further into training (day 5+), the US pause showed lower magnitudes than those observed on earlier days (−13 ± 30.3 vs −28.8 ± 30.4 Hz, resp., p=0.0018, MWU). Furthermore, on a trial-by-trial basis, 4 out of 41 recordings with a US pause and at least 20% CRs showed deeper pauses in trials with worse behavior (p=0.0017, bootstrap with 500 repetitions; *Figure 6H*). Note that, as with the US peak, US pauses were observed less frequently in the second dataset (23.5%). These findings establish a transient post-US suppressive response that is observed most reliably in conditions associated with climbing fiber-driven complex spikes in Purkinje cells.

## US pauses are followed by fast excitatory responses in IpN neurons after conditioning

In addition to the US peak and pause responses, 83 IpN cells in the first dataset showed a rapid excitatory response that peaked at 46 ± 7 ms and showed magnitudes of 61.4 ± 28.3 Hz (39.4%; *Figure 7A*, arrow), which we refer to as the second US peak. While the amplitudes of the first and second US peaks were correlated trial-by-trial in IpN neurons (*Figure 7D*), their relation to the behavioral paradigm was different. Like the first US peak, the second peak was more prevalent in recordings with poor conditioned behavior (26 ± 32 Hz, vs 13 ± 26 Hz, p=0.025, MWU; *Figure 7E*), but it did not show a steady decline over the course of training days (r = 0.102, p=0.13, n = 224, Spearman). To the contrary, across recordings in which CR performance was poor, the second US peak actually grew stronger over the course of conditioning days (r = 0.287, p=0.0007, n = 138, Spearman), with maximum peak amplitudes 30–60 ms post-US of 16 ± 28.3 Hz on training days 1–4 (n = 121), and 46.5 ± 33 Hz on later training days (n = 46, p=0.0001, MWU; *Figure 7B*). Similarly, spike rate velocity was lower during the second US peak on earlier training days compared to later ones (0.55 ± 0.32 vs 0.77 ± 0.45 Hz/ms, p=0.0072, MWU; *Figure 7C*).

Consistent with the observation that US-related responses in IpN neurons tended to occur during recordings in which CR performance was poor (*Figure 6*; *Figure 7E*), we found that in the second dataset (in which CR performance was very high) only 9 cells (8.8%) showed a first US peak response, only 24 (23.5%) showed a US pause, and only 11 (10.8%) showed a second US peak (CR amplitude: 31.3 ± 15.9 vs 44.9 ± 16.6% eyelid closure, p=0.021, MWU; *Figure 7E*). To examine US responses of IpN neurons in the second dataset, we interspersed US-only trials (n = 13 ± 2) during recordings of 25 cells (24.5%), of which 13 showed CS-US facilitation (*Figure 7—figure supplement 1A,B*). Indeed, both the first US peak (10/25 cells, 40%) and US pause (10/25, 40%) showed a higher prevalence in US-only trials than in paired trials, but the second US peak was the most substantial, occurring in 18 cells (72%), and outranking the first US peak in terms of amplitude (94.6 ± 82.9 vs 27.8 ± 48.7 Hz, p=0.0012, MWU). Interestingly, it was especially pronounced in US-only trials in the 12 recordings that showed clear CS pauses and CS-US facilitation in paired trials (*Figure 7—figure supplement 1A–C*). In fact, there was a remarkable similarity between the pause and subsequent rapid excitation observed after the US in US-only trials, and the pause and subsequent rapid excitation observed after the CS in paired trials (*Figure 7F,G*), hinting at the possibility of a similar underlying mechanism.

## Discussion

By quantifying neuronal responses in the cerebellar interpositus nucleus (IpN) during and after Pavlovian eyeblink conditioning, the present study sheds light on the question of how cerebellar cortical and extra-cortical afferents are integrated at the level of the cerebellar nuclei. Trial-by-trial correlations and their temporal distribution across the trial timespan, as well as optogenetic abolition of CRs, evince and characterize the causal relation between IpN facilitation and conditioned eyelid-behavior. Our findings, consistent across two independent datasets, together characterize cerebellar cortico-nuclear integration as a multi-faceted process involving the impact of common afferents. The data suggest that the dynamic modulation of activity in the IpN during the generation of conditioned eyelid movements can be affected through a learned Purkinje cell simple spike suppression on the one hand, and through climbing fiber mediated Purkinje cell complex spike activity, possibly in tandem with activation of mossy fiber and climbing fiber collaterals, on the other (*Figure 8*).

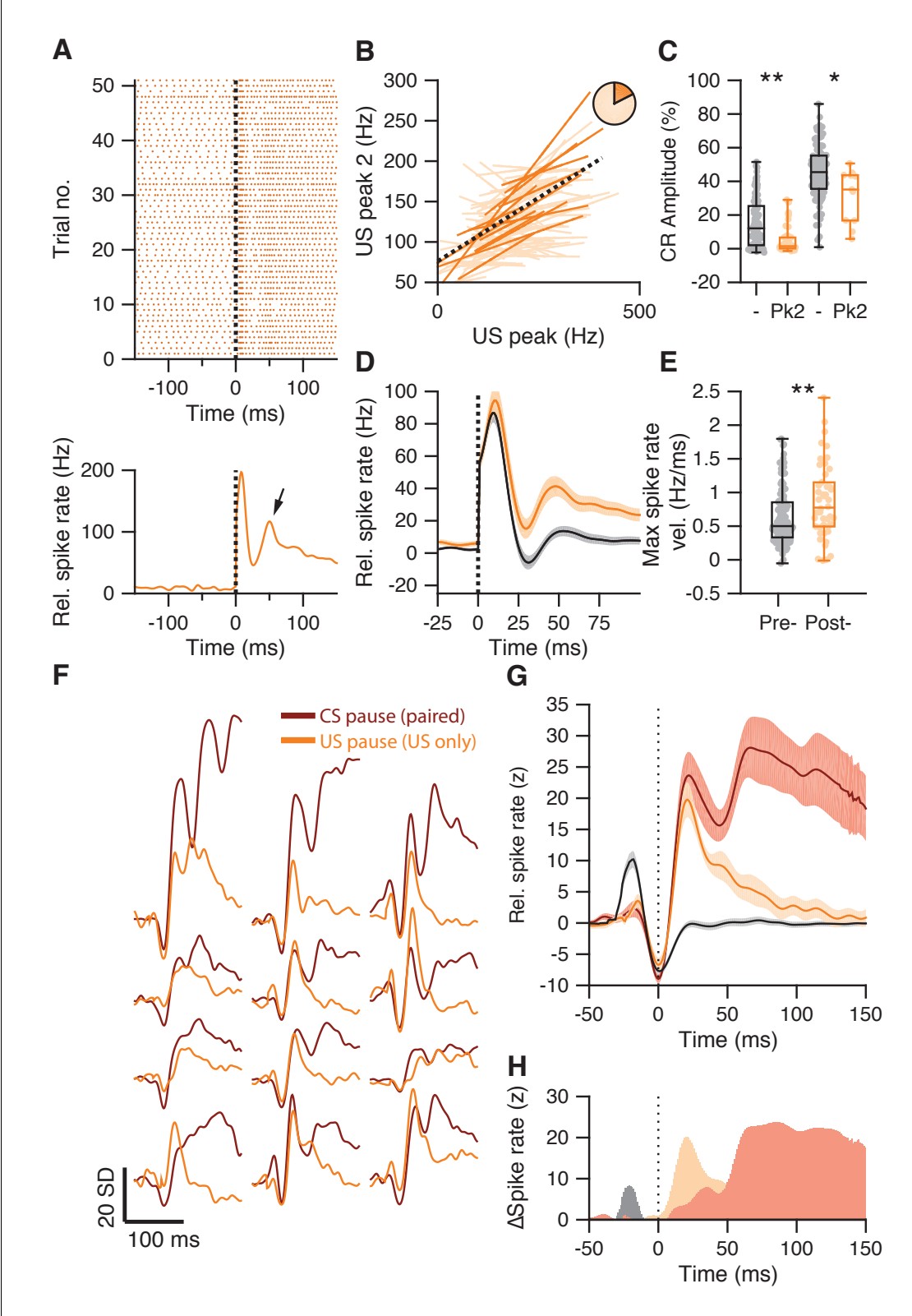

**Figure 7.** Second US peak response akin to rapid post-CS pause excitation. (**A**) Raster plot (upper panel) and average spike trace (lower panel) of an example IpN cell showing a second US peak (arrow). (**B**) Significant (plain orange) and non-significant (light orange) trial-by-trial correlation lines for cells showing both a first and a second US peak (n = 83). Pie chart shows the proportion of significant cells, black dotted line shows fit from a linear mixed model, integrating significant cells. (**C**) IpNs with a second US peak (Pk2) show lower average CR amplitudes at US onset than IpNs without (-), in both

*Figure 7 continued on next page*

*Figure 7 continued*

the first dataset (left boxplot pair; p<0.0001), as well as in the second dataset (right boxplot pair; p=0.021). (D) Average traces with SEM of spike activity relative to baseline for IpNs recorded without conditioned behavior on days 1–4 (black), and those recorded without conditioned behavior after day 4 (yellow). (E) Maximum spike rate velocity of the second US peak was significantly higher in recordings without CR behavior on day five and over (post-) compared to earlier recordings without behavior (pre-; p=0.0072). (F) Average spike traces of paired trials, aligned by CS pause minima (brown), and of US-only trials from the corresponding IpN cell, aligned by US pause (yellow) minima, for 12 recordings from the second dataset. The traces were standardized by the activity −150 to −50 ms relative to the pause minima. (G) Averages of the CS pause (brown) and US pause (yellow) traces in F, with SEM. For reference, US pause-aligned traces from paired trials of IpNs from the first dataset, with US pauses, without CR behavior, and from day 1–4, are shown in black. (H) The difference between US-responses in US-only trials after training (yellow) and US-responses in paired trials early in training before CR behavior is acquired (gray) highlight the absence of the first US peak and the substantial presence of a second peak after the US pause, in well-trained animals. Additionally, the difference between the CS pause-aligned traces from paired trials (brown) and the US-pause aligned traces from US-only trials of the same IpN set (yellow) suggests the two profiles only start to diverge substantially approx. 50 ms after the pause response.
DOI: https://doi.org/10.7554/eLife.28132.014

The following figure supplements are available for figure 7:

**Figure supplement 1.** IpN responses in paired trials and US-only trials, after optimal conditioning.
DOI: https://doi.org/10.7554/eLife.28132.015

**Figure supplement 2.** Rapid post-CS pause excitation relates to broader subsequent excitation and to CR behavior.
DOI: https://doi.org/10.7554/eLife.28132.016

## Facilitation drives the learned response

About one third of the US-responding cells recorded across the IpN over the course of conditioning showed a significant change in spiking activity within the CS-US interval, with a facilitation to suppression ratio of 2.6 (60:23). In the second dataset, recording from optimally conditioned mice in eyelid-controlling IpN regions, 85% showed CS-US spike modulation, with a facilitation to suppression ratio of 4.1 (70:17). Although facilitation has also been shown to predominate over suppression in previous work (*McCormick et al., 1982*; *Berthier and Moore, 1990*; *Gould and Steinmetz, 1996*; *Freeman and Nicholson, 2000*; *Gruart et al., 2000*; *Choi and Moore, 2003*; *Green and Arenos, 2007*; *Halverson et al., 2010*; *Heiney et al., 2014b*), its profile and relation to eyelid behavior has been not been consistently characterized as leading (for posterior IpN, see *Gruart et al., 2000*; *Delgado-García and Gruart, 2005*) and its role has been hypothesized to be only a facilitator, rather than a main driver, of CRs (*Delgado-García and Gruart, 2005*). In all, the comprehensive trial-by-trial correlational analysis reported here confirms the leading nature of facilitation in IpN cells, in terms of the temporal profiles of modulation as well as the temporal distribution of spike-eyelid correlations. Moreover, optogenetic stimulation of Purkinje cells, which was shown to suppress the activity of cerebellar nuclei neurons (*Witter et al., 2013*; *Canto et al., 2016*), effectively abolished CRs. One potential complication could have been the obstruction of CRs by the elicitation of twitches, but the inhibitory effect in the IpN that results from our protocol would only have been able to induce twitches at its offset (*Witter et al., 2013*), which fell outside the CS-US interval. The opposite potential caveat could be a sudden drop in muscle tone and a resulting immobility. However, this should have been reflected in reduced natural fluctuations of eyelid behavior compared to baseline, for which there was no indication in our data. Thus, while it was shown previously that activation of IpN via optogenetic suppression of Purkinje cells is sufficient to elicit eyelid behavior (*Heiney et al., 2014a*), we here expand on pharmacological evidence for the necessity of facilitation in the IpN for the expression of conditioned eyelid behavior (*Freeman et al., 2005*; *Ohyama et al., 2006*; *Bracha et al., 1994*; *Heiney et al., 2014b*), by inactivating IpN cells, only during the CS-US interval, through optogenetic stimulation of their inhibitory Purkinje cell input.

IpN neurons showing CR-related suppression of activity have previously been reported in *Berthier and Moore (1990)* and *Gruart et al., 2000*, and were hypothesized to facilitate conditioned behavior through inactivation of antagonistic muscles (*Gruart et al., 2000*). However, whereas the facilitation IpN cells that we recorded showed predictive trial-by-trial correlations to conditioned eyelid responses, to the modest extent that the suppressive IpN neurons showed correlations, they did only to simultaneous or past, but not future, behavior, across both datasets. While the overall minor presence of suppression in IpN neurons, and their modest, reflective correlations may be indicative of a minimal role in driving behavior, hinting perhaps at sensory feedback, the relaxation of antagonistic muscles is a secondary, passive means of facilitating CRs; its benefit would

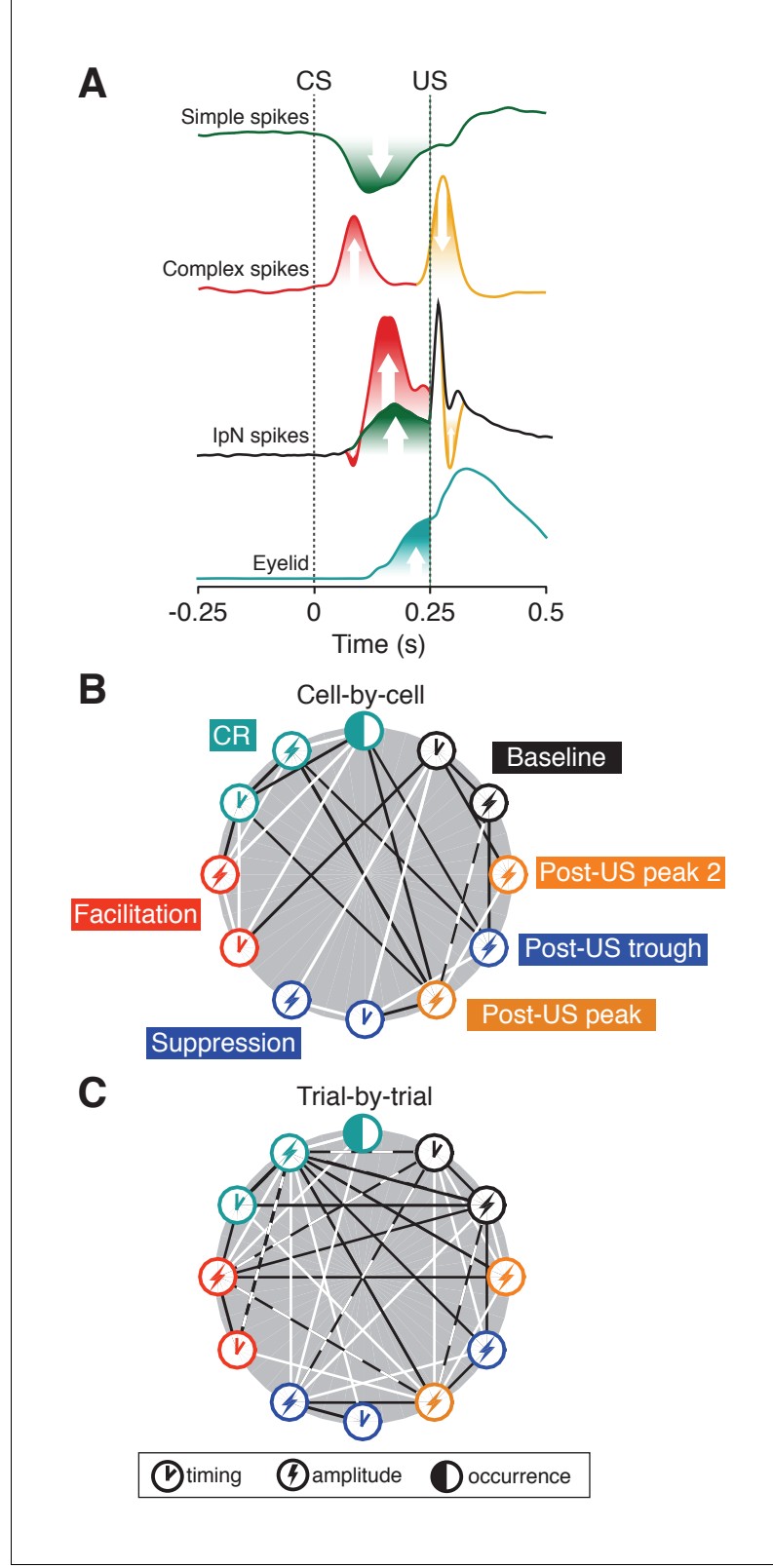

**Figure 8.** Eyelid-related modulation in cerebellar cortex and nuclei. (**A**) Mean spike traces for simple and complex spikes in Purkinje cells and IpN spike show how eyelid-related cerebellar cortical and nuclear modulation relate after training. On the one hand, simple spike suppression translates relatively straightforwardly to facilitation in the IpN. Stimulus-related complex spike input leaves a transient spike trough, with the well-timed CS-related complex
*Figure 8 continued on next page*

*Figure 8 continued*
spike input showing a tendency to be followed by strong rebound-like excitation. Extra-cortical afferents underlie the oft observed transient post-US excitatory response, and presumably interact with the other modulatory components. The timing component in the baseline represents the coefficient of variation (CV). (B) Cell-by-cell correlations between occurrence, amplitude, and timing of the different eyelid behavior and IpN spiking components. Black lines indicate negative correlations, white lines indicate positive ones. (C) Trial-by-trial correlations between the components also labeled in B, with lines indicating the presence of a significant number of cells showing significant correlations of the corresponding direction. For each parameter pair, recordings were included only if they were considered to show the associated phenomena. Dashed white and black lines indicate that both positive and negative correlations were significantly present among cells.
DOI: https://doi.org/10.7554/eLife.28132.017

in part comprise a reduced energy cost to effect behavior, and may therefore escape proper detection in a correlational analysis focusing only on eyelid kinematics. Alternatively, as our electrodes might have preferentially targeted the large projection neurons, the suppression cells might also represent smaller inhibitory interneurons of the nuclei that in fact could serve to further shape the activity of the facilitation cells, smoothing the CR.

## Stereotyped pause-excitation responses in the IpN

Many IpN neurons responded at the onset of the conditioned eyelid movement with a characteristic transient pause, which was followed by a period of increased activity. The transient IpN spike pauses observed across datasets strikingly mirror the CS- and US-related complex spike responses found in eyelid-related Purkinje cells (*Ohmae and Medina, 2015*; *ten Brinke et al., 2015* and *Figure 4*). The translation from complex spike input to a pause response in the IpN is supported by the findings that climbing fiber activation can trigger a prominent inhibition in the cerebellar nuclei (*Hoebeek et al., 2010*; *Bengtsson et al., 2011*; *Lu et al., 2016*; *Tang et al., 2016*) and that the strength of this inhibition can be related to the synchrony of the complex spikes (*De Zeeuw et al., 2011*; *Tang et al., 2016*).

The excitatory response of IpN neurons is normally attributed to learning-related suppression of Purkinje cell activity, plasticity of the excitatory mossy fiber to IpN connection, or a combination of the two (*Medina et al., 2000*; *Ohyama et al., 2006*; *Longley and Yeo, 2014*; *Freeman, 2015*). To these potential mechanisms, our results add one more: In the first dataset, the fastest and strongest cases of excitatory modulation in the IpN neurons occurred right after the CS-related spike pause (*Figure 4*). Moreover, in the second dataset, a similar pattern of pause followed by strong excitation was observed in US-alone trials, which are known to trigger climbing fiber-driven complex spikes in Purkinje cells (*Hesslow, 1994a*; *Mostofi et al., 2010*). This CS-related spike pause is likely to have a direct impact on the subsequent spike modulation, as the modulation amplitudes and velocity were significantly higher in the cells with a CS-related spike pause (*Figure 5*). This characteristic profile fits well with the notion of rebound depolarization in the cerebellar nuclei (*Hoebeek et al., 2010*; *Bengtsson et al., 2011*). This phenomenon is well established in vitro (*Jahnsen, 1986*; *Llinás and Mühlethaler, 1988*; *Aizenman and Linden, 1999*; *Tadayonnejad et al., 2009*), and can be elicited through electrical (cerebellar nuclei and inferior olive, *Hoebeek et al., 2010*; inferior olive and skin, *Bengtsson et al., 2011*) and optogenetic (inferior olive, *Lu et al., 2016*; cerebellar nuclei, *Witter et al., 2013*) stimulation. In particular, rebound-driven behavioral responses can be elicited by synchronized nuclear inhibition lasting as little as 25 ms (*Witter et al., 2013*). *Bengtsson et al., 2011* and *Hoebeek et al. (2010)* report that particularly synchronized complex spike input seems fit to induce substantial rebound excitation in the cerebellar nuclei. Nevertheless, there are also stimulation studies showing less convincing rebound (*Chaumont et al., 2013*), and its spontaneous occurrence in vitro or in vivo seems negligible (*Alviña et al., 2008*; *Reato et al., 2016*). Given the dependence of several models of cerebellar learning on the ability of the cerebellar nuclei to use rebound depolarization to trigger mechanisms of plasticity (e.g., *Kistler et al., 1999*; *Steuber et al., 2007*; *Wetmore et al., 2008*; *De Zeeuw et al., 2011*), the data here presented are particularly relevant in that they offer some evidence for an affirmative answer to the open question of whether rebound may occur during learning (*Reato et al., 2016*).

## IpN neurons respond widely to unconditioned stimuli

We observed that many IpN neurons responded strongly to presentation of the US (*Figure 6*). The rapid post-US excitatory response had an average latency of 10.6 ms (25th-75th percentile: 7–12 ms), which allows for both mossy fiber and climbing fiber collateral activation as possible pathways for driving the response (*Berthier and Moore, 1990*; *Cody and Richardson, 1979*; *Mostofi et al., 2010*). *Mostofi et al. (2010)* point at latencies < 6 ms for mossy fiber activation as recorded in the cerebellar cortex in awake rabbits in response to peri-ocular electrical stimulation, and latencies > 9 ms for climbing fiber responses, which is in line with work in cats by *Cody and Richardson (1979)*.

The fact that we used air puff instead of electrical stimulation, which introduces a delay, and that our subjects were mice, which may introduce a small reduction in transmission time, confound proper appraisal of the pathway underlying the US peak based on its latency. Nevertheless, several properties of the US peak provide some clues as to its origin. On the one hand, one could argue that the robust and ubiquitous manifestation of the US peak across the IpN suggests mossy fiber collaterals are a more likely source than climbing fiber collaterals. IpN cells that showed eyelid-related facilitation constitute a selective subgroup of cells in the IpN, and neurons showing US pauses were more commonly observed in less deeply located IpN neurons, which agrees with the notion that the anterior IpN is the main target region for eyelid-related Purkinje cell output (*Yeo et al., 1985a1985*; *Steinmetz et al., 1992*; *Bracha et al., 1994*; *Krupa and Thompson, 1997*; *De Zeeuw and Yeo, 2005*; *Freeman et al., 2005*; *Heiney et al., 2014b*). If climbing fiber collaterals would be distributed as widely throughout the IpN as the US peak was observed to be, this would almost certainly mean they are not constrained within eyelid-related olivocerebellar modules. Therefore, mossy fiber input, which is not assumed to adhere to such a modular organization and forms a more robust input to the cerebellar nuclei (e.g. *Person and Raman, 2010*), may more appropriately fit the observed manifestation of the US peak. Moreover, the reduction of US peak responses over the course of conditioning may well reflect the reduced impact of the air puff US due to protection of the eyelid, and a resultant reduced mossy fiber signal. On the other hand, there is research to suggest that climbing fiber collateral input may be more robust (*Van der Want et al., 1989*; *Hesslow, 1994b*; *De Zeeuw and Yeo, 2005*) than the notion of 'sparse collaterals' suggests (*Chan-Palay, 1977*). Moreover, stimulation of the inferior olive results in a consistent excitatory response in the cerebellar nuclei (*Kitai et al., 1977*; *Hoebeek et al., 2010*), and spontaneous complex spikes elicit a co-occurrence of excitatory and inhibitory responses in the nuclei 24% of the time (*Blenkinsop and Lang, 2011*), which is not very different from the percentage we observed (32%). The cell- and trial-wide correlations suggesting weaker US-peaks with better CRs may also be explained in terms of a changing level of nucleo-olivary inhibition (*Medina et al., 2002*), which would agree with a contribution of the climbing fiber collaterals (*De Zeeuw et al., 1997*). Moreover, the interesting occurrence of weaker US peaks in trials where mice opened their eyes further than baseline, agrees with Purkinje cell US-complex spike data from *ten Brinke et al. (2015)*, underlining another similarity between the US peak and climbing fiber activity. Thus, while these arguments suggest it is likely that climbing fibers contribute to the US peak response, its broad prevalence across the IpN suggests mossy fibers are also involved.

The question remains as to why the US-related responses were substantially more widespread than the CS-related responses. Considering the strong convergence of Purkinje cells onto IpN neurons (approx. 40:1) (*Person and Raman, 2011*), only a relatively small portion of the IpN neurons might evoke functional behavioral responses. As the large pool of remaining IpN neurons may show US-responses that relay general sensory information of the face, including that of the areas surrounding the eyelid region (*Koekkoek et al., 2002*), or feed into cognitive processes (*Wagner et al., 2017*; *Sokolov et al., 2017*), there may also be a substantial amount of redundant, yet detectable, spike responses. This would be consistent with recent findings that dense population coding during eyeblink conditioning also occurs at the cerebellar input stage, i.e. in the granule cell layer (*D'Angelo et al., 2009*; *Giovannucci et al., 2017*), which may partly result from nucleocortical feedback originating in the IpN (*Houck and Person, 2015*; *Gao et al., 2016*).

## Functional implications

In the present study, we have detailed the learning-related dynamic modulation of cerebellar IpN cells and evaluated it against the previously observed modulation of Purkinje cells of the cerebellar

cortex by which they are inhibited during Pavlovian eyeblink conditioning. Through the use of detailed trial-by-trial correlations with conditioned eyelid behavior as well as computational modeling and optogenetic confirmation we established the necessity of facilitation of IpN cells for the production of conditioned responses. Moreover, we have uncovered a well-timed imprint of presumptive complex spike activity in the IpN and a subsequent strong excitation. Thus, possibly in interaction with direct input to the IpN, climbing fiber input to the cerebellar cortex could provide an additional mechanism through which spike modulation may be elicited in the IpN to fine-tune the timing of CRs (*De Zeeuw and Ten Brinke, 2015*). These results underline the relevance of olivary climbing fibers beyond their conventional role of providing teaching signals, highlighting a unique attribute of the olivocerebellar system compared to other networks, which generally segregate the signals that are used for inducing plasticity during learning from signals that are used for performance during memory recall.

## Materials and methods

### Surgery

Subjects were 12–20 week-old wild-type C57Bl/6 mice (n = 25), housed individually with food and water ad libitum in a normal (n = 22, first dataset) or reversed (n = 6, second dataset) 12:12 light/dark cycle. For the optogenetics experiment, we used L7cre-Ai27 mice (n = 3) that express channelrhodopsin-2 in Purkinje cells. The experiments were approved by the institutional animal welfare committees (Erasmus MC, Rotterdam, The Netherlands and Baylor College of Medicine, Houston, USA). Mice were anesthetized with an isoflurane +oxygen mixture (5% for induction, 2% for maintenance) and body temperature was kept constant at 37°C. After fixation in a standard mouse stereotaxic alignment system (Stoelting Co., Wood Dale IL, USA), the scalp was opened to expose the skull. Membranous tissue was cleared, and the bone was surgically prepared with Optibond prime and adhesive (Kerr, Bioggio, Switzerland). A small brass pedestal was attached to the skull with Charisma (Heraeus Kulzer, Armonk NY, USA), using an xyz-manipulator, allowing for fixation to a head bar at right angles during training and electrophysiology. For the craniotomy performed after training, skin and muscle tissue was cleared from the left half of the occipital bone, where, after applying a local analgesic (bupivacainehydrochloride 2.5 mg ml-1), a roughly 1 mm wide craniotomy was performed, 1.5 mm from midline. A small rim of Charisma was made around the craniotomy and anti-inflammatory (Dexamethasone 4 mg ml-1) solution was applied inside, after which the chamber was closed with a very low viscosity silicone elastomer sealant (Kwik-cast, World Precision Instruments, Sarasota FL, USA).

### Eyeblink conditioning

Two days after surgery, mice were head-fixed to a brass bar suspended over a cylindrical treadmill (*Chettih et al., 2011*) and placed in the electrophysiology set-up, contained in a light-isolated Faraday cage. A first habituation session consisted of 30–45 min during which no stimuli were presented. During a second and third habituation session on consecutive days 10 CS-only trials were presented to allow the mice to get used to the green LED light and to acquire a baseline measurement. Eight mice were subjected to a training paradigm for five days, before the electrophysiology phase, receiving 200 paired trials daily, with an inter-trial interval of 10 ± 2 s, amounting to approximately half an hour per session. In the remaining 11 mice, electrophysiology was performed from the start of training onwards, restricting the amount of paired trials to 250 per day to ensure comparability between mice across days.

The CS was a 260 ms green LED light, placed ~7 cm in front of the mouse. The US was a 10 ms corneal air-puff at 40 psi delivered through a 27.5-gauge needle tip positioned 5–10 mm from the left eye, co-terminating with the CS, which amounts to a CS-US interval of 250 ms. National Instruments NI-PXI (National Instruments, Austin TX, USA) processors and a low-noise Axon CNS Digidata 1440a acquisition system (AutoMate Scientific Inc., Berkely CA, USA) were used to trigger and keep track of stimuli whilst capturing data. Eyelid movements were recorded with a 250-fps camera (scA640-120gm, Basler, Ahrensburg, Germany). Eyeblink conditioning for the second dataset was performed as described in *Heiney et al. (2014b)*.

For each recording, eyelid traces were normalized to the full blink range, which consisted of the minimal resting baseline value reflecting the *open eye* position as established visually during the experiment, and the mean of the UR peak values reflecting the *closed eye* position. The traces were smoothed using a 2nd degree Savitzky-Golay method with a span of 10 ms. An iterative Grubbs' outlier detection test (α = 0.05) on trial baseline standard deviations was used to remove trials that had an unstable baseline. Additionally, trials were removed if the eye was not at least halfway open. Next, CR amplitude was quantified as the maximum eyelid position within the CS-US interval relative to the trial baseline position, expressed as percentage of full blink range. Trials were considered to contain a CR when eyelid closure exceeded 5% from the baseline mean within the CS-US interval. CR onset was determined as the first time point of a continuous positive eyelid velocity leading to up to the fifth percentile of the amplitude from baseline to CR peak.

## Electrophysiology

Neurons for the first dataset were recorded with glass capillaries (Ø=2 mm, Harvard Apparatus, Holliston MA, USA) that were heated and pulled to obtain a 2–5 μm tip, and filled with a 2M NaCl solution. The electrode was stereotactically lowered into the IpN using an electrode holder that was positioned at a 40° angle in the caudal direction on the sagittal plane and controlled by a manipulator system (Luigs and Neumann SM7, Germany). The obtained electrical signal was pre-amplified with a computer-controlled microelectrode amplifier (Axon CNS, MultiClamp 700B, AutoMate Scientific, Inc. US) and digitized at 20 kHz using the Axon Digidata acquisition system. Neurons for the second dataset were recorded as described in *Heiney et al., (2014b)*. The IpN contains a high density of neurons that are deeper than and well separated from cortical layers, and almost always show clear, single unit spike activity. Upon encountering cells with responses to the CS and/or US, and verifying a stable recording, animals were subjected to blocks of paired trials. The trial-by-trial spike-eyelid correlations in *ten Brinke et al., (2015)* showed significant r values ranging from 0.34 to 0.73, effect sizes that require 12–65 trials to be detected with the ideal statistical power of 80%. Since the same correlations for IpN neurons may be higher since their activity is located more closely to eyelid behavior, and since there was no previous information suggesting expected effect sizes for the other neuronal responses, we included all recordings with at least 10 trials. Neurons were recorded for up to 7 days, and 1% Cholera toxin B subunit (CTB, C9903, Sigma-Aldrich) was used to replace 2M saline in the electrode on the last day of electrophysiology. Eventually, 50 nL of CTB was injected after the last recoding. Electrophysiological recordings were band-pass filtered at 150–6000 Hz and spikes were thresholded in MATLAB (RRID:SCR_001622, The Mathworks, Natick MA, USA) using custom-written code and SpikeTrain (Neurasmus, Rotterdam, The Netherlands). Spike density functions (referred to as average spike traces) were computed for all trials by convolving binary 1 ms-bin vectors containing spike occurrences with a Gaussian kernel with a 5 ms width.

## Microstimulation mapping of eyeblink-controlling regions in IpN

Functional mapping of the CN was performed using 80-μm-diameter platinum iridium monopolar electrodes (100 KΩ; Alpha Omega) as in *Heiney et al., (2014b)*. Electrodes were positioned using stereotaxic coordinates relative to a mark on the cement. Mapping was mostly confined to the anterior interpositus (IpN), dorsolateral hump (DLH), and lateral nucleus (LN). Electrodes were advanced in steps of 100 μm, and currents in the range of 1–15 μA (200 ms pulse trains; 250 μs biphasic pulses; 500 Hz) were systematically tested to identify the threshold for evoking movement. The eyeblink-controlling region of IpN was defined as the location at which discrete and sustained eyelid closure could be evoked with low currents (<5 μA). This region was subsequently targeted for electrophysiological recordings.

## Optogenetics

For optogenetic inhibition of the cerebellar nuclei, we used the L7cre-Ai27 mouse line, in which ChR2 is specifically expressed in Purkinje cells upon the expression of Pcp2-cre. An optic cannula fiber (Ø105 μm, 0.22NA, 2 mm, Thorlabs) was vertically implanted over the IpN (AP 2.5 mm, ML 2 mm) and attached to the skull. In addition, a brass pedestal was attached to the skull between bregma and lambda. Animals were allowed to recover from surgery for 2 weeks. After eyeblink conditioning, mice were head-fixed on a set up that is equipped with an optogenetic blue light source

(wave length 470 nm, M470F3, Thorlabs) and a high-power light driver (M00283732, Thorlabs). In a 50-trial session, 10 trials were randomly presented together with a 5V optic stimulus that gave 5 ms pulses at 100 Hz for the duration of the CS (260 ms), thus terminating at CS and US offset.

## Spike modulation in the last 200 ms of the CS-US interval

Spike rate suppression and facilitation in the CS-US interval (*Figure 2*), relative to a 500 ms baseline period, were determined separately. We chose to determine average spike modulation of cells in terms of absolute instead of relative change from the average baseline firing frequency, as we wanted to not process the data further away from actual firing frequency than necessary. Moreover, absolute change was not different across firing frequencies in terms of facilitation (r = 0.025, p=0.68, n = 270, Spearman), and absolute suppression was only slightly deeper in faster firing cells (r = −0.131, p=0.0308, n = 270, Spearman), whereas relative mean modulation was more clearly linked to average firing frequency (facilitation: r = −0.378, p<0.0001; suppression: r = −0.211, p=0.0005; n = 270, Spearman). Thus, expressing spike modulation in absolute terms thus avoids the use of a modulation criterion that is implicitly looser for cells with lower firing frequencies. We considered cells to carry CS-US facilitation if they showed an average increase of at least **5 Hz** above baseline mean in the last 200 ms of the CS-US interval, after subtracting the mean above-baseline activity within the baseline itself. Similarly, cells showing an activity of at least **5 Hz** below baseline mean, correcting for below-baseline activity within the baseline itself, were considered suppressive. For both types of modulation, this threshold roughly corresponds to 3 SDs among the incidentally negatively modulating cells, i.e. cells where the baseline showed more facilitation/suppression than the CS-US interval. Duration of modulation was calculated as the sum of ms-bins exceeding 3 baseline SDs in the last 200 ms of the CS-US interval; spike firing needed to exceed this threshold for at least 10 ms during the CS-US interval.

## CS-responses

To see how IpN neurons may reflect CS-related Purkinje cell complex spikes, rapid shifts in spike frequency were assessed between 50 and 125 ms post-CS in average spike traces that were standardized to baseline. A rapid drop exceeding 4.5 SDs was considered to be a CS pause if it was followed by a peak of at least 2 SDs. Similarly, an upstroke of 4.5 SDs was labeled a CS peak if it was followed by a drop of at least 2 SDs.

## US-responses

First, the presence of US peak and pause responses in individual cells were quantified on the basis of their average spike traces, standardized to the 500 ms baseline. The time windows used to quantify the first US peak and pause (*Figure 6*), and the second US peak (*Figure 7*) were 1–30 ms, 16–45 ms, and 31–60 ms post-US, respectively. A first US peak was identified if it exceeded either 5 or 2 baseline SDs from the average of the last 50 ms of the CS-US interval, with the latter case also requiring a 5 SD difference from baseline. This way, cells that showed high facilitation in the CS-US interval were not ignored simply because they did not spare enough room for the spike increase required to meet the first criterion. Pauses were recognized if they exceeded 5 baseline SDs below the final 50 ms of the CS-US interval, and were at least 2 SDs below baseline. Alternatively, they could be 2 SDs below the end of the CS-US interval and exceed 5 SDs from baseline. This means pauses that did not dip below baseline level were not recognized as such, as it is difficult to ascertain whether they are pauses, or just the space between two peaks. Finally, second peaks were considered significant if they were still at least 2 SDs above the last 50 ms of the CS-US interval as well as 5 SDs above baseline. For both the pause and the second peak, the subsequent turnaround had to regress at least 2 SDs back to baseline, so as to avoid the inclusion of blunt, unpeak-like movements in the average spike traces. For trial-by-trial analyses, absolute firing frequency was used, taking the maximum (or, for pauses, minimum) values within 30 ms time-windows centered at the average peak- or pause-time.

## Simple spike-based model of IpN neurons

Using Python software (RRID:SCR_008394), A total of 26 IpN neurons were modeled (*Figure 5*) using the parameters shown in *Figure 5—source data 1*, with each modeled IpN neuron incorporating

the simple spike modulation of a Purkinje cell from the dataset in *ten Brinke et al. (2015)*. For each modeled trial of IpN modulation, 30 simple spike patterns were used as input to reflect the reported convergence rate of 20–50 Purkinje cells to 1 nuclear cell (*Person and Raman, 2011*). It is not known how similar or dissimilar simple spike modulation of Purkinje cells projecting to the same nuclear cell is. We here opted to take the same reference Purkinje cell for each of the thirty simple spike patterns used to model IpN activity. To avoid reusing trials, we expanded the Purkinje cell simple spike data. For each Purkinje cell, inter-spike interval distribution (ISI) characteristics (mean, SD, skewness, kurtosis) were determined across trials for each 1 ms bin in the trial timespan. By iteratively sampling a random ISI from the local ISI distribution, trials of simple spike activity are created that carry the same firing and modulation characteristics as the original Purkinje cell. The resulting set of 900 trials of simple spike activity was fed to the model, resulting in 30 trials of IpN spike activity. Spike density functions were subsequently computed for each IpN cell and parameters across the combined data were compared with actual IpN recordings.

## Statistics

Significance tests were performed using MATLAB. Unless otherwise specified, mean and standard deviation were used to report central tendency and measure of deviation. Correlations between two continuous variables were made using Pearson's r if there were at least 30 data points and data was normally distributed for both variables, after a Grubbs' test for outliers ($\alpha$ = 0.05); otherwise, Spearman's rho was used. Group differences were assessed with Mann-Whitney U tests. For combined analysis of multiple cells, mixed-effects linear regression was performed, with random intercepts and slopes for cells included based on likelihood ratio tests. Visual inspection of residuals safeguarded adequate normality and homoscedasticity, and helped identify extreme outliers (never more than four). Bootstrap tests were used to determine the probability for a certain number of cells to be significant across a cell population, by assessing significance across 500 random datasets with similar amounts of cells and similar amounts of trials. Correlation matrices were used to explore the temporal distribution of the correlations between spike modulation and behavior (*Figure 2*, *Figure 2—figure supplement 1*). The procedure is identical to the one explained in *ten Brinke et al., (2015)*. In short, the trial timespan is divided up in 20 ms bins. Across trials, the average eyelid value within each bin is correlated to the average spike rate value within each bin, resulting in a 100 × 100 matrix of r-values, given the 2 s trial window. Matrices for multiple cells were subsequently compiled by taking the average r-values across matrices, after nullifying the r-values of the sign opposite to the sign of interest. Note that this constitutes a methodological tool used to assess the temporal distribution of correlations, and not to quantify their significance.

## Acknowledgements

Authors are grateful to Erika Goedknegt and Elise Haasdijk for technical assistance. We thank the Dutch Organization for Medical Sciences (ZonMw; CIDZ), Life Sciences (ALW; ZG, CIDZ), the ERC-advanced, CEREBNET and C7 programs of the European Community (CIDZ), and NIH grants R01MH093727, RF1MH114269 and R21AA025572 (JFM) for their financial support.

## Additional information

### Funding

| Funder | Grant reference number | Author |
| --- | --- | --- |
| National Institutes of Health | R01MH093727 | Javier F Medina |
| National Institutes of Health | RF1MH114269 | Javier F Medina |
| National Institutes of Health | R21AA025572 | Javier F Medina |
| Nederlandse Organisatie voor Wetenschappelijk Onderzoek | | Zhenyu Gao Chris I De Zeeuw |
| European Research Council | | Chris I De Zeeuw |

The funders had no role in study design, data collection and interpretation, or the decision to submit the work for publication.

## Author contributions
Michiel M ten Brinke, Conceptualization, Formal analysis, Writing—original draft; Shane A Heiney, Martina Proietti-Onori, Conceptualization, Data curation, Formal analysis, Writing—review and editing; Xiaolu Wang, Data curation, Formal analysis, Writing—review and editing; Henk-Jan Boele, Conceptualization, Writing—review and editing; Jacob Bakermans, Formal analysis; Javier F Medina, Zhenyu Gao, Chris I De Zeeuw, Conceptualization, Supervision, Writing—review and editing

## Author ORCIDs
Michiel M ten Brinke  https://orcid.org/0000-0002-9478-1586
Shane A Heiney  http://orcid.org/0000-0001-9706-2133
Xiaolu Wang  http://orcid.org/0000-0002-2353-5775
Jacob Bakermans  http://orcid.org/0000-0003-1645-2645
Zhenyu Gao  http://orcid.org/0000-0002-4979-2366

## Ethics
Animal experimentation: The experiments were approved by the institutional animal welfare committee (Erasmus MC, Rotterdam, The Netherlands). All surgery was performed under isoflurane anaesthesia, and every effort was made to minimize suffering.

## Decision letter and Author response
Decision letter https://doi.org/10.7554/eLife.28132.027
Author response https://doi.org/10.7554/eLife.28132.028

# Additional files

## Supplementary files
• Source data 1. Source data for IpN datasets. For 51 variables relevant to the analyses in this paper, the values for each trial for each cell are contained in a data structure in the attached MATLAB. mat-file. The structure is named 'ipn', and contains 51 instances; for each variable instance, there is a 'name' field, describing the variable, and a 'values' field, containing the data. The 'values' field contains a cell array, with each cell containing the values of the selected variable for all trials of an actual cell. The two IpN datasets central to this study are combined, with the first 270 cells denoting the first dataset, and the remaining 102 cells denoting the second dataset.
DOI: https://doi.org/10.7554/eLife.28132.018
• Supplementary file 1. Summary statistics of the linear mixed model analyses in *Figures 2*, *6* and *7*.
DOI: https://doi.org/10.7554/eLife.28132.019
• Transparent reporting form
DOI: https://doi.org/10.7554/eLife.28132.020

## Major datasets
The following dataset was generated:

| Author(s) | Year | Dataset title | Dataset URL | Database, license, and accessibility information |
|---|---|---|---|---|
| Ten Brinke MM, Heiney SA, Wang X, Proietti-Onori M, Boele HJ, Bakermans J, Medina JF, Gao Z, De Zeeuw CI | 2017 | Eyelid behavior and spike activity of cerebellar interpositus nucleus neurons during eyeblink conditioning in awake behaving mice | http://dx.doi.org/10.6080/K0B85697 | Publicly available at the Collaborative Research in Computational Neuroscience (http://crcns.org/). |

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
