## [Decision Letter]

Thank you for submitting your article "Dynamics of Cerebellar Nuclei Neurons during Pavlovian Eyeblink Conditioning" for consideration by *eLife*. Your article has been reviewed by three peer reviewers, and the evaluation has been overseen by Naoshige Uchida as the Reviewing Editor and Richard Ivry as the Senior Editor. The following individuals involved in review of your submission have agreed to reveal their identity: John Freeman (Reviewer #1); David Herzfeld (Reviewer #3).

The reviewers have discussed the reviews with one another and the Reviewing Editor has drafted this decision to help you prepare a revised submission.

Summary:

Eyeblink conditioning has been a great model for studying the neural mechanism underlying classical conditioning. Although the cerebellum has long been implicated in eyeblink conditioning, how neurons in in the deep cerebellar nuclei (DCN) and those in the cerebellar cortex regulate eyeblink conditioning is yet to be determined. The present study provides a comprehensive analysis of the firing of DNC neurons (in particular, those in the interposed nucleus [IpN]) in relation with learning and Purkinje cell (PC) activities. Most notably:

1) The authors demonstrate a tight relationship between PC complex spikes and firing pauses of DCN neuron. This work builds on the work of Eric Lang (presence of complex spikes in DCN activity) and Javier Medina (presence of complex spikes due to the conditioned stimulus).

2) The authors show a trial-by-trial correlation between the activity of "facilitatory" DCN cells and eyeblink behavior.

3) Optogenetic activation of PCs caused inhibition of DCN neurons and reduced the conditioned response (eyeblink), supporting the necessity of normal IpN activity for conditioned behavior (although the enthusiasm of this result varied among the reviewers).

The reviewers thought that this work provides an important set of results that advances our understanding of the contributions of different neuron types in the cerebellum in eyeblink conditioning. Nonetheless, the reviewers noted some issues with respect to their interpretation and presentation of the results. We therefore would like to see the authors' response to the following points:

Essential points:

1) Most of the analyses rely on discrete classifications of neurons based on their response types, and do not provide a comprehensive view on the diversity of responses in each population. Furthermore, the main results such as correlation between neural activities and behaviors were obtained using a relatively small fraction of neurons that exhibited particular responses. The reviewers are generally concerned on these points, and suggest that the authors provide additional analyses or discussions to remedy these issues, taking into account the diversity of observed response types in the data set. More detailed comments regarding this are as follows:

1a) The authors have recorded from a large set of diverse DCN neurons. They are generally subdivided by a particular response property and then analyzed within those subdivisions. This subdivision can be confusing or problematic, as the criteria often change from experiment to experiment. For example, in Figure 1, facilitation/ suppression cells are defined as having at least 5 Hz up/down modulation. First, it would be helpful if the authors show a histogram of the population, as it is not clear that these are distinct populations. Second, later in the manuscript (subsection “CS-related transient spike trough reflects acquired CS-related complex spike response”), the amount of facilitation is compared between this population and a separate population which was defined in a different way. Furthermore, in some cases, the numbers of neurons in each category is so small (e.g. Figure 4) that it remains unclear how many different response types there are, and how much each type really contributes to the behavior.

1b) The proportion of neurons with learning-relevant activity is lower than found in many of the previous studies cited. Only 29% showed facilitation and 17 of those showed positive correlations with the CR. This raises a concern regarding whether or not the recordings were from neurons in an eyelid control area of the IpN. A related concern is that some of the previous studies examining IpN activity used electrical stimulation through the recording electrode to elicit eyelid movement to determine whether the neurons recorded from that electrode were in an eyelid control area of the IpN. There are problems associated with this method such as activating fibers of passage, but the authors should note in the paper that some of the IpN neurons could have been controlling CRs for other muscles that happen to be highly correlated with eyelid closure.

1c) Modeling needs to be performed carefully, taking into account the scarcity of some of the key response types. In other words, a major challenge for the interpretation of the current results is how to weight the contributions of neurons exhibiting particular response types as indicated in Figure 9. For example, the authors stress the importance of the neurons showing a pause in spiking early in the CS period, but there were only 7 of these neurons. It appears that the study yielded sufficient data to model the interactions described in Figure 9, taking into account all of the various contributors in the population, weighted by their prevalence, magnitude of neuronal modulation, and strength of correlation with CRs.

1d) In the Materials and methods section, the descriptions of selection criteria for e.g. suppression vs. troughs are confusing. This is partly because the authors have specific terminologies that are not always obvious (for instance, it was not clear that there a fundamental difference in the meaning of 'suppression' vs. 'trough'). The authors should reference the specific figures/ experiments to which they refer for each section. Please use consistent descriptive terminology indicating response profiles associate with CS/US/complex spikes.

2) The comparison of the IpN neurons with predicted responses based on models of summed PC inputs (Figure 5) is very interesting. However, the authors in general seem to discount the role of non-PC inputs to DCN, specifically mossy fibers. For instance, the larger facilitation in neurons that show a CS-trough is interpreted as being due to rebound potentiation. How can enhanced MF plasticity onto those neurons (resulting from stronger CS modulation) be ruled out?

3) It is unclear about the putative relationship between PC complex spikes and the peak activity shown in Figure 4. From Blenkinsop and Lang (2011), it would seem that if there is facilitation in the DCN following a PC complex spike, it should be brief compared to the depression (representing short-latency climbing fiber excitatory collaterals rather than complex spike related inhibition from the PCs). However, the duration of the excitatory peak in Figure 4 seems comparable to those provided in the early part of the same figure. Are these really complex spike mediated responses? Consider that IpN cell 2 in Figure 4 seems to show suppression with a presumptive complex spike following the US (opposite of the bursting relationship after CS).

In essence, there is no data to distinguish clearly between complex spike vs. climbing fiber afferent regulating DCN responses. The reviewers were concerned that the impact statement (as well as parts of the Discussion) goes beyond the data. Please revise the impact statement and other related descriptions to reflect this concern.

4) The Materials and methods section for the optogenetic experiments is missing.

5) Simultaneous activity of presynaptic PCs likely inhibits IpN cells well below their basal firing rate. Depending on the duration of stimulation, this could affect the kinematics of the unconditioned response. Do the authors see any changes in the kinematics of the eye blink for stimulation versus pre/early-training in which there is no CR?

6) It is unclear whether the results of this study are applicable to cerebral cortical interactions with the thalamus as suggested in the first paragraph of the Introduction and last paragraph of the Discussion. The reviewers thought that the nature of connectivity is very different, with cerebellar cortical output being exclusively inhibitory, for example. The reviewers thought that the Introduction is oversold and largely irrelevant. The reviewers suggest removing the discussion on cortical-subcortical interactions.

7) The author's trial-by-trial correlation analysis could be strengthened. The authors calculated correlations between 20ms bins of spike responses and behavior across trials. This analysis inherently assumes that each 20ms bin is independent, which is not the case. It would be good to use the actual time-dependent spiking responses to predict the CR behavior in time. A potential analysis: average rate responses of a neuron across trials in which a CR occurred (e.g., day 7+). For the same trial period determine the learned CR as the average difference between the eye position trace after learning and the position traces before learning. Now do a correlational analysis between these two average timeseries. What is the optimal lag/lead across neurons? How well does the facilitation timeseries predict the learned CR response (in time)?

8) The emergence of facilitation and suppression shown in Figure 1 occurs much more rapidly than does the emergence of behavior. This seems to disagree with the correlation results and optogenetic stimulation in the rest of the paper. If IpN facilitation is both necessary and sufficient to drive eye-blink behavior why does the end of Day 5 (which has approximately the same facilitation as Day 7) have significantly less CRs? Perhaps this is due to averaging spike responses across the entire CS-US period in Figure 1. If the authors plotted the 20ms bin average at the ideal lag (80ms), would this look qualitatively different that the average responses plotted in Figure 1 (e.g., emerging slower)?

---

## [Author Response]

Essential points:1) Most of the analyses rely on discrete classifications of neurons based on their response types, and do not provide a comprehensive view on the diversity of responses in each population. Furthermore, the main results such as correlation between neural activities and behaviors were obtained using a relatively small fraction of neurons that exhibited particular responses. The reviewers are generally concerned on these points, and suggest that the authors provide additional analyses or discussions to remedy these issues, taking into account the diversity of observed response types in the data set. More detailed comments regarding this are as follows:

We understand the general concern that some of the numbers of neurons showing particular responses were low. We have therefore added more cells and analyses in addition to expanding our discussion of the findings. These additions would not have been possible without substantial efforts from Dr. Shane Heiney and Dr. Javier Medina at Baylor College of Medicine in Houston, who have been added as co-authors. As a consequence, a more comprehensive evaluation is now provided and all issues relating to the interpretation of the diversity of conditioning-associated responses, particularly those with modest prevalence, are addressed.

1a) The authors have recorded from a large set of diverse DCN neurons. They are generally subdivided by a particular response property and then analyzed within those subdivisions. This subdivision can be confusing or problematic, as the criteria often change from experiment to experiment. For example, in Figure 1, facilitation/ suppression cells are defined as having at least 5 Hz up/down modulation. First, it would be helpful if the authors show a histogram of the population, as it is not clear that these are distinct populations. Second, later in the manuscript (subsection “CS-related transient spike trough reflects acquired CS-related complex spike response”), the amount of facilitation is compared between this population and a separate population which was defined in a different way. Furthermore, in some cases, the numbers of neurons in each category is so small (e.g. Figure 4) that it remains unclear how many different response types there are, and how much each type really contributes to the behavior.

We appreciate the point that the neuronal subdivision was not always clear, and have now made an effort to streamline our explanations in both Results and Materials and methods. Our choice to use a 5 Hz-threshold is based on a number of considerations. First, our preference was to not process spike modulation data further away from actual firing frequency than necessary. Second, when comparing baseline firing frequency to relative CS-US modulation expressed as Hz, it is clear that this approach does not introduce a baseline frequency dependent selection bias. In contrast, when we express it as a percentage in relation to baseline frequency, it does induce a bias in that neurons with low baseline frequency are preferred. For selection bias, see Author response image 1 5 Hz-threshold identifies modulating cells evenly (red plots), whereas percentage threshold shows preference towards low frequency cells (orange plots, general population top correlation plots, subsets of facilitation cells see bottom plot). Moreover, when looking at the distribution of suppression and facilitation (blue and red histograms, resp.), the negative values indicate occasions when there was more modulation in the baseline than in the CS-US interval; the one-sided 3 SD thresholds for these incidental magnitudes (dotted blue and red lines) both are in close agreement with a 5 Hz threshold (dashed black lines). Thus, given its intuitive straightforwardness, the fact that it does not introduce bias across neurons showing different baseline firing frequencies, and its statistical appropriateness, we decided on using a 5 Hz threshold. This rationale is now explained in the Materials and methods section.

With regards to the comparison in the subsection “CS-related transient spike trough reflects acquired CS-related complex spike response”, facilitation is not computed any differently in the two groups that are compared. Rather, all facilitation cells were split into a group with cells that also showed a putatively CS-complex spike related spike trough, and one with cells that did not. We have now rephrased this part to be clearer. Moreover, with the additional data, we now provide a more comprehensive dataset, upping the numbers for the different categories of neuronal responses. Lastly, we now provide a visualized classification of these different groups in Figure 1—figure supplement 1.

1b) The proportion of neurons with learning-relevant activity is lower than found in many of the previous studies cited. Only 29% showed facilitation and 17 of those showed positive correlations with the CR. This raises a concern regarding whether or not the recordings were from neurons in an eyelid control area of the IpN. A related concern is that some of the previous studies examining IpN activity used electrical stimulation through the recording electrode to elicit eyelid movement to determine whether the neurons recorded from that electrode were in an eyelid control area of the IpN. There are problems associated with this method such as activating fibers of passage, but the authors should note in the paper that some of the IpN neurons could have been controlling CRs for other muscles that happen to be highly correlated with eyelid closure.

The proportion of neurons with learning-relevant activity is dependent on the selection criteria and paradigms. Berthier and Moore (1990) observed similar numbers of 50 out of 165 cells showing CR-related facilitation (30.3%) in trained animals. Moreover, when we use the same criteria as Halverson, Lee, and Freeman (2010), we would have 41% facilitation cells. For this study we decided to use the 5 Hz selection threshold for the reasons elaborated in the previous point.

Our original approach, recording throughout the IpN, is one reason why the strong prevalence of almost immediate excitation after the US implies the ubiquity of whichever collaterals are responsible for them (climbing fibers seem to play a substantial role, given our data and the compatibility between the bimodal latencies of US-complex spikes in Mostofi et al. (2010) and US-peak responses in Berthier and Moore, (1990)). Nevertheless, we concede the downside that some of our IpN neurons could have been controlling other highly correlated muscles than the eyelid. We therefore addressed this concern by incorporating a second dataset, which allowed us to corroborate our findings in recordings that were demonstrably eyelid-controlling, by virtue of micro-stimulation. As pointed out by the reviewer this approach, although not perfect either, does indeed provide a higher percentage of facilitation cells (see new Figure 1—figure supplement 1). We now explain this comparison and highlight this reasoning in the Discussion.

1c) Modeling needs to be performed carefully, taking into account the scarcity of some of the key response types. In other words, a major challenge for the interpretation of the current results is how to weight the contributions of neurons exhibiting particular response types as indicated in Figure 9. For example, the authors stress the importance of the neurons showing a pause in spiking early in the CS period, but there were only 7 of these neurons. It appears that the study yielded sufficient data to model the interactions described in Figure 9, taking into account all of the various contributors in the population, weighted by their prevalence, magnitude of neuronal modulation, and strength of correlation with CRs.

We agree that in the original submission, the scarcity of key response types made it difficult to appraise their relevance, and indeed the relevance of their interactions as described in Figure 9. In the current version of the manuscript, we have alleviated this concern through the addition of substantial data obtained in optimally conditioned mice. This has greatly sharpened the image in terms of the manifestation of neuronal responses, e.g. by drastically increasing the number of cells showing CS pause responses (Figure 4), and in contrast by leaving the CS peak response wholly uncorroborated (and thus relegated to Figure 4—figure supplement 1). The layered pie-charts in Figure 1—figure supplement 1 now provide an extensive overview of how the different response types co-occur, also distinguishing between cells that do or don’t show significant spike-eyelid correlations. Finally, we have now updated the modeled diagram in Figure 9 (now Figure 8) to include the second dataset, thereby greatly strengthening the visualized interactions.

1d) In the Materials and methods section, the descriptions of selection criteria for e.g. suppression vs. troughs are confusing. This is partly because the authors have specific terminologies that are not always obvious (for instance, it was not clear that there a fundamental difference in the meaning of 'suppression' vs. 'trough'). The authors should reference the specific figures/ experiments to which they refer for each section. Please use consistent descriptive terminology indicating response profiles associate with CS/US/complex spikes.

We have now included references to specific figures/experiments in the Materials and methods, and, avoiding the terms “trough” and “facilitatory”, now use the following terminology:

CS-US facilitationIpN spike facilitation in the CS-US interval (Figure 2).CS-US suppressionIpN spike suppression in the CS-US interval (Figure 2).CS pauseTransient IpN spike pause reflecting the CS-related complex spike (Figure 4).US peakIpN spike burst occurring 0-30 ms after US onset (Figure 6).US pauseTransient IpN spike pause reflecting the US-related complex spike at 15-30 ms after US onset (Figure 6).Second US peakRapid spike facilitation following complex spike-related US pause (Figure 7).

2) The comparison of the IpN neurons with predicted responses based on models of summed PC inputs (Figure 5) is very interesting. However, the authors in general seem to discount the role of non-PC inputs to DCN, specifically mossy fibers. For instance, the larger facilitation in neurons that show a CS-trough is interpreted as being due to rebound potentiation. How can enhanced MF plasticity onto those neurons (resulting from stronger CS modulation) be ruled out?

With the additional data that is now included in the manuscript (particularly in Figure 8), it has become apparent that, after training, a US-only trial can elicit a spike trough and subsequent rapid excitatory response with a profile that quite accurately fits that of the trough and rapid excitation observed after the CS. This suggests that a contribution of potentiated mossy fiber input is not necessary per se. Still, we do not rule out MF plasticity, and would in fact agree that it could play a role in the facilitation observed in IpN neurons showing the CS-related spike trough. For MF-DCN synapses to potentiate, their activation likely needs to be combined with a post-inhibitory rebound depolarization (Pugh and Raman, 2006, 2008), which we propose could be provided by the synchronized complex spike input likely underlying the CS pause. Still, if the larger facilitation we report is indeed partly due to potentiated mossy fiber collateral synapses, the CS signals conveyed through them from the pontine nuclei would have to have a curious temporal profile, with a maximum right after the CS-related spike trough. While this is a rather tough requirement for MF-DCN LTP to explain the larger IpN facilitation we report, the temporal profiles of pontine activity reflecting 300 ms tone CSs does offer some plausibility (Figure 3 in Clark, Gohl, and Lavond, 1997). The more straightforward version of rebound excitation, not dependent on LTP at MF-DCN synapses, also involves the confluence of excitation and inhibition, which leads to concomitant activation of GABAA and mGluRs that together drive subsequent rebound firing (Zheng and Raman, 2011) with a temporal profile that readily fits the temporal profile of the larger IpN facilitation we observed. While the inhibition may be provided by the possibly synchronized CS-related complex spikes, the requisite excitation could be provided by mossy and/or climbing fiber collaterals. We now make reference to the potential role of both mossy and climbing fiber collaterals in the Discussion.

3) It is unclear about the putative relationship between PC complex spikes and the peak activity shown in Figure 4. From Blenkinsop and Lang (2011), it would seem that if there is facilitation in the DCN following a PC complex spike, it should be brief compared to the depression (representing short-latency climbing fiber excitatory collaterals rather than complex spike related inhibition from the PCs). However, the duration of the excitatory peak in Figure 4 seems comparable to those provided in the early part of the same figure. Are these really complex spike mediated responses? Consider that IpN cell 2 in Figure 4 seems to show suppression with a presumptive complex spike following the US (opposite of the bursting relationship after CS).In essence, there is no data to distinguish clearly between complex spike vs. climbing fiber afferent regulating DCN responses. The reviewers were concerned that the impact statement (as well as parts of the Discussion) goes beyond the data. Please revise the impact statement and other related descriptions to reflect this concern.

Indeed, we agree that our datasets did not differentiate between complex spike vs climbing fiber afferent induced facilitation. Therefore, we are now more cautious to draw any conclusion in the Discussion. Interestingly enough, CS related peak responses were rarely observed in the second dataset added to the study, suggesting it is a rare phenomenon within eyelid-controlling regions of the IpN. We therefore relegated this response to Figure 4—figure supplement 1, now only presenting it as a curiosity. Thus, on the broader point of overreach in the impact statement and parts of the Discussion regarding a potential contribution of climbing fiber collaterals, we agree that this assertion was not based on sufficiently robust data in our study and falls short of accurately conveying the actual significance of our findings. As a consequence, together with the remarks in point 6, we have now rephrased the impact statement to better reflect our findings, and toned down our extrapolations.

4) The Materials and methods section for the optogenetic experiments is missing.

We have now added the missing section.

5) Simultaneous activity of presynaptic PCs likely inhibits IpN cells well below their basal firing rate. Depending on the duration of stimulation, this could affect the kinematics of the unconditioned response. Do the authors see any changes in the kinematics of the eye blink for stimulation versus pre/early-training in which there is no CR?

To explore the possibility that optogenetic stimulation could affect the US responses, we compared the US responses with optogenetic inhibition with those in the trails with no CR. In essence, we looked at the difference in UR kinematics between trials with a CR and those without a CR, where the No-CR trials were due to optogenetic stimulation in one dataset, and due to a natural absence of behavior in the other dataset (IpN recordings). From the latter dataset, we took all recordings (n = 12, shown in panel A) with at least 30 trials, with conditioned behavior (>20%), but no more than 90% CRs, and with an average CR amplitude that is within the range of the average CR amplitudes in the three optogenetic experiments (shown in panel B). For each recording/session, we next extracted the difference between the averages of CR and No CR trials (panel C). Finally, for each recording/session, we determined the z-scores, by performing a bootstrap analysis of 500 repetitions (panel D). The bootstrap specifics (per recording/session): for each repetition, we randomly redistributed trials across the CR/No-CR groups, retaining the original Ns per group, and computed the difference as in C. The z-scores for the actual differences in C were then computed by subtracting for each data-point the corresponding mean of the bootstrapped dataset and subsequently dividing by the corresponding standard deviation of the bootstrapped dataset.

**Author response image 2. respfig2:** 

Whereas on average the results indicate no significant difference between CR and No-CR trials anywhere in the UR phase in the IpN dataset, we do find an effect in the optogenetics data. All three experiments are suggestive of a tendency of the mice to keep their eyelid more closed between 100-300 ms post-US (= post-stimulation offset), with session’s average passing the 2 SD line, and another even passing the 3 SD line. The time window within which the average z-score trace for the optogenetic data was above 2 SDs is indicated by the shaded pink area in panels B and D. The delay between stimulation offset and this time window of substantial difference is roughly similar to that reported in Witter et al. (2013; Figure 5). Thus, the data here presented suggest a behavioral effect due to post-stimulation rebound that is consistent with previous work using a similar optogenetic methodology (Witter et al., 2013). Given the already substantial size of the manuscript and this observation’s relative distance from the central aims of this study, we would prefer not to include it in the paper, unless the reviewers/Editors disagree.

6) It is unclear whether the results of this study are applicable to cerebral cortical interactions with the thalamus as suggested in the first paragraph of the Introduction and last paragraph of the Discussion. The reviewers thought that the nature of connectivity is very different, with cerebellar cortical output being exclusively inhibitory, for example. The reviewers thought that the Introduction is oversold and largely irrelevant. The reviewers suggest removing the discussion on cortical-subcortical interactions.

We have now removed most of the cortical-subcortical angle, and rewrote the Introduction’s first and Discussion’s last paragraph to more tightly reflect the scope of our study.

7) The author's trial-by-trial correlation analysis could be strengthened. The authors calculated correlations between 20ms bins of spike responses and behavior across trials. This analysis inherently assumes that each 20ms bin is independent, which is not the case. It would be good to use the actual time-dependent spiking responses to predict the CR behavior in time. A potential analysis: average rate responses of a neuron across trials in which a CR occurred (e.g., day 7+). For the same trial period determine the learned CR as the average difference between the eye position trace after learning and the position traces before learning. Now do a correlational analysis between these two average timeseries. What is the optimal lag/lead across neurons? How well does the facilitation timeseries predict the learned CR response (in time)?

The relation between the amplitudes of spiking and eyelid response can be analyzed in two orthogonal ways: I) by correlating across time points within individual trials or the average across trials, as now implemented in Figure 2—figure supplement 1 as per the reviewers’ suggestion; or II) by correlating across trials at individual time points (Figure 2) or the average across time points within the CS-US interval (Figure 2), which is our original approach. Whereas the data in the former across-time points analysis concern a discrete sampling of a continuous process (one could interpolate between time data points), the data from the across-trial analysis actually concerns discrete, independent measurements (from different trials), and is therefore suitable for statistical inferences, as we draw from single analyses in Figure 2.

The temporal dynamics of the relationship between spike and eyelid modulation can be explored in terms of two aspects that are also orthogonal: I) the degree of relatedness at different times in the trial timespan; and II) the degree of relatedness at different temporal offsets, i.e. the lead/lag relationship. The suggested temporal cross-correlation method explores the latter, but gives no information about which part or parts of the trial timespan carry the relatedness between spikes and eyelid. By contrast, the correlation matrix analysis, as shown in Figure 2, readily allows for the integrated evaluation of both aspects: relatedness can be assessed at different times in the trial timespan (I) by moving along the diagonal parallel to the dashed white line, and at different lead/lag relationships (II) by moving along the perpendicular/orthogonal diagonal. Thus, different temporal components of the spike-eyelid relationship can be identified, at different times and with different lead/lag relationships, such as the two hotspots in the Purkinje cell correlation matrix in ten Brinke et al. (2015; Figure 2). The early hotspot in that matrix, relating the short CS-complex spike response to the full CR, and the relatively broad hotspot at an adjacent time in Figure 6 are consistent with the notion of a CS-related complex spike input triggering rebound-excitation in the IpN. Thus, the correlation matrix approach allows for but does not assume independent components across time-bins, whereas the suggested temporal cross-correlation approach actually assumes dependence across time points. This can be problematic when using spike activity with multiple components that may or may not be interdependent (as we see in our dataset), because with across-timepoint correlations, these components are conflated.

Nevertheless, we agree that temporal cross-correlation could indeed strengthen our analysis, for two reasons. First, in the event there is little meaningful across-trial amplitude variation, it can still provide information about the lead/lag relationship between the conditioned spike and eyelid responses. Second, when applied to each trial individually instead of the average across trials, the resultant dataset can give a 95% confidence interval of the optimal lead/lag relationship, based on independent measurements. This analysis is now included in the manuscript (Figure 2—figure supplement 1), and shows lead/lag relationships that are consistent with our original analyses.

8) The emergence of facilitation and suppression shown in Figure 1 occurs much more rapidly than does the emergence of behavior. This seems to disagree with the correlation results and optogenetic stimulation in the rest of the paper. If IpN facilitation is both necessary and sufficient to drive eye-blink behavior why does the end of Day 5 (which has approximately the same facilitation as Day 7) have significantly less CRs? Perhaps this is due to averaging spike responses across the entire CS-US period in Figure 1. If the authors plotted the 20ms bin average at the ideal lag (80ms), would this look qualitatively different that the average responses plotted in Figure 1 (e.g., emerging slower)?

The most straightforward conceptualization for simultaneous emergence of spike modulation and behavior would assume a uniform one-on-one relation, meaning the magnitude of spike modulation is representative of its population and proportional to the corresponding eyelid behavior. Based on our data, we think there could well be a threshold that needs to be reached before spike modulation starts manifesting as CRs. This could be in terms of magnitude of modulation, and/or number of IpN cells being recruited.

The finding that spike modulation occurring ahead of eyelid behavior is supported by previous work, such as Freeman and Nicholson (2000), Halverson, Lee, and Freeman (2010), and McCormick and Thompson (1984).

As per the reviewers’ suggestion, we have plotted the facilitation data from Figure 1 in 20ms bins, in four different ways, shown in Author response image 3. Red and blue distinguish between relative firing rates expressed in Hz or in Z-scores (resp.), and plain and light color shades distinguish between the full CS-US interval and the CS-US interval up until 170ms (resp.). These two ranges assume no lag or an 80ms lag, respectively, which together with the eyelid velocity analysis, and the new temporal cross-correlational analyses, is the outer limit (the optimal lag likely lies between the two). Note that the recordings included here all met the facilitation criterion, like the corresponding data in Figure 1. While among these different visualizations one may agree with CR development better than others, there does not seem to be a game-changing difference with regards to the matter of asynchronous development of spike and eyelid modulation.

**Author response image 3. respfig3:** 

Altogether, rather than suboptimal averaging of spike responses, we would suggest, as presented in the manuscript, that the development of the spike modulation ahead of the behavioral expression reflects the actual development of the conditioning process.